# The role of the PZP domain of AF10 in acute leukemia driven by AF10 translocations

Brianna J. Klein[1,7], Anagha Deshpande[2,7], Khan L. Cox[3], Fan Xuan [4], Mohamad Zandian [1], Karina Barbosa [2], Sujita Khanal[2], Qiong Tong[1], Yi Zhang [1], Pan Zhang[2], Amit Sinha[5], Stefan K. Bohlander [6], Xiaobing Shi [4], Hong Wen [4], Michael G. Poirier[3], Aniruddha J. Deshpande [2✉] & Tatiana G. Kutateladze [1✉]

Chromosomal translocations of the AF10 (or MLLT10) gene are frequently found in acute leukemias. Here, we show that the PZP domain of AF10 (AF10$_{PZP}$), which is consistently impaired or deleted in leukemogenic AF10 translocations, plays a critical role in blocking malignant transformation. Incorporation of functional AF10$_{PZP}$ into the leukemogenic CALM-AF10 fusion prevents the transforming activity of the fusion in bone marrow-derived hematopoietic stem and progenitor cells in vitro and in vivo and abrogates CALM-AF10-mediated leukemogenesis in vivo. Crystallographic, biochemical and mutagenesis studies reveal that AF10$_{PZP}$ binds to the nucleosome core particle through multivalent contacts with the histone H3 tail and DNA and associates with chromatin in cells, colocalizing with active methylation marks and discriminating against the repressive H3K27me3 mark. AF10$_{PZP}$ promotes nuclear localization of CALM-AF10 and is required for association with chromatin. Our data indicate that the disruption of AF10$_{PZP}$ function in the CALM-AF10 fusion directly leads to transformation, whereas the inclusion of AF10$_{PZP}$ downregulates *Hoxa* genes and reverses cellular transformation. Our findings highlight the molecular mechanism by which AF10 targets chromatin and suggest a model for the AF10$_{PZP}$-dependent CALM-AF10-mediated leukemogenesis.

[1] Department of Pharmacology, University of Colorado School of Medicine, Aurora, CO, USA. [2] Tumor Initiation and Maintenance Program, National Cancer Institute-Designated Cancer Center, Sanford Burnham Prebys Medical Discovery Institute, La Jolla, CA, USA. [3] Department of Physics, Ohio State University, Columbus, OH, USA. [4] Center for Epigenetics, Van Andel Research Institute, Grand Rapids, MI, USA. [5] Basepair Inc, New York, NY, USA. [6] Leukaemia and Blood Cancer Research Unit, Department of Molecular Medicine and Pathology, University of Auckland, Auckland, New Zealand. [7] These authors contributed equally: Brianna J. Klein, Anagha Deshpande. ✉email: adeshpande@sbpdiscovery.org; tatiana.kutateladze@cuanschutz.edu

Human AF10 (or mixed-lineage leukemia translocated to 10 (MLLT10)) is essential in hematopoiesis and implicated in blood cancers. Chromosomal translocations involving the AF10 gene are frequently found in acute lymphoblastic leukemia (ALL) and acute myeloid leukemia (AML)[1–7]. These aggressive forms of leukemia affect predominantly children and young adults and are characterized by poor survival rates[8,9]. At least seven translocation partners of AF10 have been identified, including the most common partners clathrin assembly lymphoid myeloid leukemia (CALM) and KMT2A. The leukemia-associated AF10 translocations are shown to dysregulate downstream signaling programs since they produce aberrantly active fusion oncoproteins.

Although AF10 represents primarily a carboxy-terminal fragment in the leukemia-associated chromosomal translocations, significant heterogeneity has been reported in AF10 fusion breakpoints. Interestingly, despite this heterogeneity, all AF10 fusion chimeras contain the C-terminal octapeptide-motif leucine zipper (OM-LZ) domain of AF10 (AF10$_{OMLZ}$) (Fig. 1a). AF10$_{OMLZ}$ is involved in the interaction with the histone methyltransferase disruptor of telomeric silencing 1-like (DOT1L), an enzyme that generates methylated H3K79 species associated with high gene expression[10–12]. Furthermore, the DOT1L recruitment to target genes and the deposition of the methylated H3K79 marks require the binding of DOT1L to AF10$_{OMLZ}$[13]. This notable and strict conservation of AF10$_{OMLZ}$ and therefore the DOT1L-binding

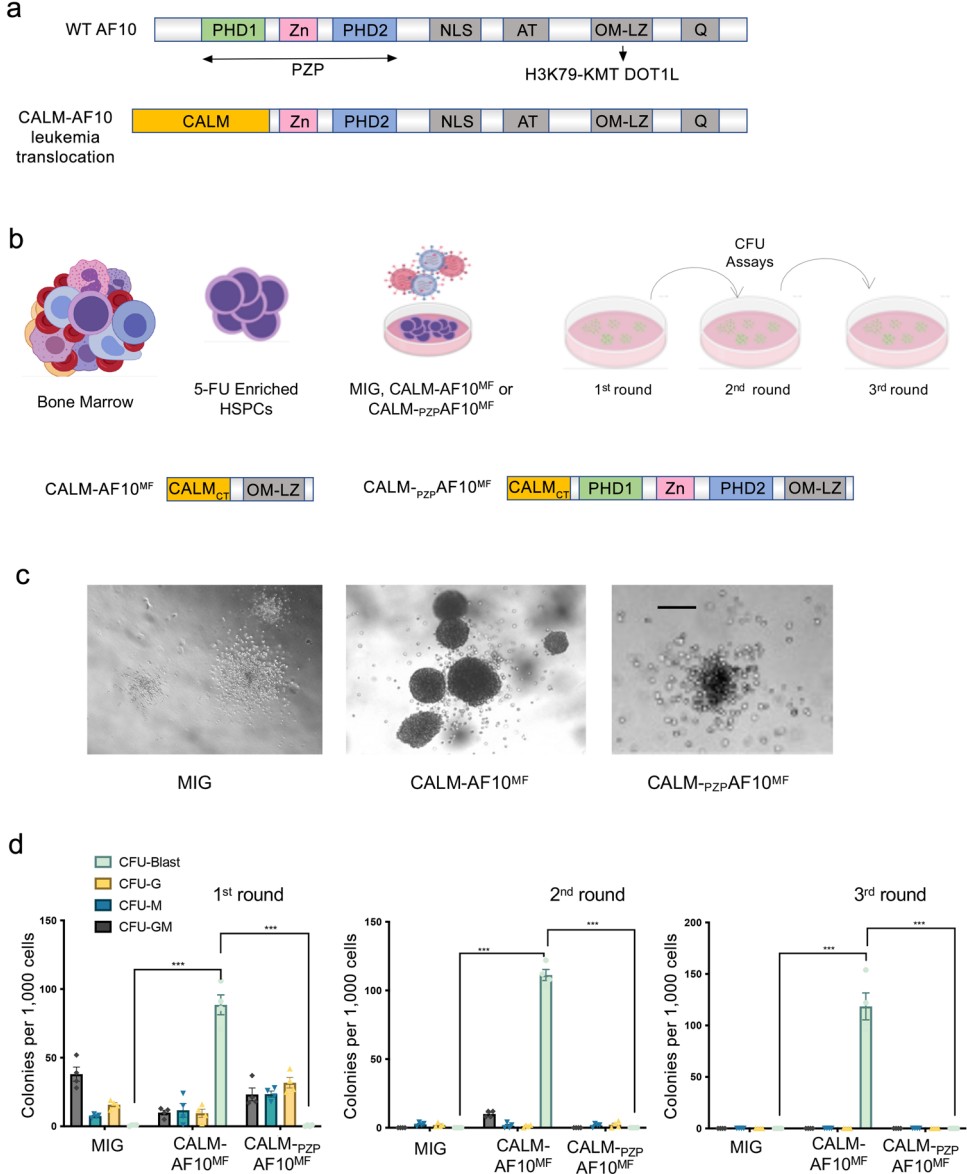

**Fig. 1 AF10$_{PZP}$ blocks transformation by the CALM-AF10 fusion. a** The architecture of WT AF10 and the CALM-AF10 fusion. AF10$_{PZP}$, comprised of PHD1, Zn-kn, and PHD2 (colored light green, pink, and light blue, respectively) is consistently disrupted in leukemogenic AF10 translocations. The H3K79-specific lysine methyltransferase (KMT) DOT1L binds to AF10$_{OMLZ}$. **b** Schematic of the CFU assay, with the architecture of CALM-AF10$^{MF}$ and CALM-$_{PZP}$AF10$^{MF}$ shown at the bottom. **c** Images of representative one-week CFU assay colonies from bone marrow-derived HSPCs transduced with each of the indicated constructs are shown at ×40 magnification. Scale bar: 100 μm. CFU assays performed in HSPCs isolated individually from 3 mice. **d** Different types of colonies from cells expressing each of the indicated plasmids are shown at 3 consecutive rounds of plating. CFU-G: colony-forming unit-granulocyte, CFU-M: colony-forming unit-macrophage, CFU-GM: colony-forming unit granulocyte monocyte, CFU-Blast: blast-like colonies. *P* values of CFU-Blast colonies (MIG vs. CALM-AF10$^{MF}$) and (CALM-AF10$^{MF}$ vs. CALM-$_{PZP}$AF10$^{MF}$) are 0.000019 for week 1, 0.00001 for week 2, and 0.000101 for week 3 (Student's *t*-test).

capability in all leukemia-associated AF10 fusions suggests a likely mechanism underlying the development of AF10-rearranged leukemias that involves the aberrant recruitment and/or stabilization of DOT1L at promoters of leukemogenic genes and constitutive activation of these genes.

The CALM-AF10 t(10;11)(p12;q14) translocation is particularly highly leukemogenic and is linked to aggressive acute leukemias. Wild type CALM (or PICALM) is involved in clathrin-mediated endocytosis, and an almost entire CALM protein, including its ENTH domain and the clathrin-binding domain, are present in the CALM-AF10 chimera, being fused to AF10 in which the first PHD finger (AF10$_{PHD1}$) is deleted. Much like other AF10 translocations, the CALM-AF10 translocation correlates with the upregulation of the proto-oncogenic *HOXA* and *MEIS1* genes. CALM-AF10 expressing cells show a local increase in H3K79 methylation on these genes but a global reduction in H3K79 methylation throughout the genome[14]. It has been proposed that CALM-AF10-mediated mislocalization of DOT1L to chromatin causes these changes in H3K79 methylation and gene expression and contributes to leukemic transformation, however, the mechanism by which DOT1L is mislocalized remains unclear. Another pressing question that needs to be addressed pertains to the role of the N-terminal PHD1-zinc-knuckle-PHD2 (PZP) domain of AF10 (AF10$_{PZP}$). AF10$_{PZP}$ is known to recognize unmodified histone H3K27 mark with methylation or acetylation of H3K27 abrogating this interaction and to oligomerize[15,16], however, whether impaired AF10$_{PZP}$ affects the transforming ability of AF10 fusions is unknown.

In this study, we describe the biological function of the PZP domain of AF10 and its critical role in inhibiting the leukemogenic activity of the CALM-AF10 translocation. We report the molecular mechanism by which AF10$_{PZP}$ recognizes a large portion of the histone H3 tail and DNA and assess the contribution of these binding events. Our data suggest that the disruption of AF10$_{PZP}$ function in oncogenic AF10 fusions leads to malignant transformation, whereas the inclusion of AF10$_{PZP}$ reverses leukemogenesis.

## Results and discussion

**AF10$_{PZP}$ prevents the transforming activity of CALM-AF10 in vitro and in vivo**. To determine the role of AF10$_{PZP}$ in the leukemogenic activity of the CALM-AF10 translocation, we modified the CALM-AF10 chimera that was reported to cause potent malignant transformation of bone marrow-derived hematopoietic stem and progenitor cells (HSPCs)[17]. This chimera consists of the C-terminal part of CALM (aa 400–648, CALM$_{CT}$), encompassing the TAD domain and NES, fused with AF10$_{OM-LZ}$ (aa 677–758) and represents the minimal fusion construct (CALM-AF10$^{MF}$) that induces transformation to the same extent as the original CALM-AF10 fusion (Fig. 1b)[17]. Because CALM-AF10$^{MF}$ does not contain AF10$_{PZP}$, we generated CALM-$_{PZP}$AF10$^{MF}$ by incorporating AF10$_{PZP}$. We then transduced bone marrow-derived HSPCs with CALM-AF10$^{MF}$, CALM-$_{PZP}$AF10$^{MF}$, or the MSCV-IRES-GFP (MIG) empty vector, purified transduced cells using a co-expressed fluorescence marker, and tested these cells in a methyl-cellulose-based semi-solid colony-forming unit (CFU) assay (Fig. 1b). As shown in Fig. 1c (middle panel), expression of CALM-AF10$^{MF}$ in bone marrow-derived HSPCs led to the formation of a large number of colonies with an undifferentiated, blast-like morphology, confirming the potent transforming ability of the CALM-AF10$^{MF}$ fusion[17]. In contrast, colonies obtained from HSPCs transduced with the CALM-$_{PZP}$AF10$^{MF}$ fusion had mostly a granulocytic (CFU-G), monocytic (CFU-M), or mixed (CFU-GM) appearance, similar to empty vector transduced cells

(Fig. 1c (left and right panels) and 1d and Suppl. Fig. 1). Furthermore, undifferentiated colonies from CALM-AF10$^{MF}$ transformed cells gave rise to the blast-like colonies in secondary and tertiary replating experiments, whereas the colonies from MIG vector or CALM-$_{PZP}$AF10$^{MF}$ transduced cells had no serial replating capacity (Fig.1d). These results suggest that the introduction of the AF10 PZP domain into the CALM-AF10 fusion abrogates the transforming ability of this chimera in vitro.

We next tested CALM-$_{PZP}$AF10$^{MF}$ in the in vivo clonogenic colony-forming unit-spleen (CFU-S) assay, in which the CALM-AF10$^{MF}$ fusion was shown to confer high CFU-S capability to bone marrow-derived HSPCs[14]. Bone marrow-derived HSPCs transduced with the CALM-AF10$^{MF}$ fusion formed a median of 100 colonies per 50,000 injected cells (Fig. 2a). In contrast, cells transduced with the CALM-$_{PZP}$AF10$^{MF}$ fusion produced only a median of 20 colonies, which is at par with the MIG vector transduced cells that produced a median of 17 colonies per 50,000 injected cells. We concluded that the incorporation of AF10$_{PZP}$ impedes the ability of the CALM-AF10$^{MF}$ fusion to form a high number of CFU-S colonies in vivo.

**The inclusion of AF10$_{PZP}$ abrogates CALM-AF10-mediated leukemogenesis in vivo**. To establish whether the inclusion of AF10$_{PZP}$ can affect the in vivo leukemogenic activity of the CALM-AF10 translocation, we injected mice ($n = 5$ mice per arm) with HSPCs transduced with either the MIG empty vector control, the CALM-AF10$^{MF}$ fusion gene, or the CALM-$_{PZP}$AF10$^{MF}$ fusion gene (Fig. 2b). While the injection of bone marrow-derived HSPCs transduced with CALM-AF10$^{MF}$ led to fully penetrant leukemias with a median of 93 days, none of the mice injected with CALM-$_{PZP}$AF10$^{MF}$ HSPCs developed disease up to 300 days post-transplantation. We next assessed whether the CALM-$_{PZP}$AF10$^{MF}$ protein, which lacks leukemogenic activity, can also block leukemogenesis via an *in trans* mechanism. We used primary leukemia cells from mice with full-blown CALM-AF10$^{MF}$-induced leukemia and transduced these cells with the CALM-$_{PZP}$AF10$^{MF}$ fusion gene. Since the cells were from a primary AML, CALM-AF10$^{MF}$ leukemia cells produced almost exclusively blast-like colonies in CFU assays. Strikingly, retroviral transduction of the CALM-$_{PZP}$AF10$^{MF}$ fusion in these leukemia cells almost completely abrogated their ability to form colonies (Fig. 2c). The ability of CALM-$_{PZP}$AF10$^{MF}$ to reverse the potent transformed phenotype of the CALM-AF10$^{MF}$ fusion indicates that AF10$_{PZP}$ has a trans-dominant tumor-suppressive function over the CALM-AF10$^{MF}$ fusion.

**Exclusion of AF10$_{PZP}$ is essential for *Hox/Meis1* activation**. The CALM-AF10 fusion is known to upregulate *HOXA* cluster genes and the HOX-cofactor *MEIS1*, which is a hallmark of this subtype of leukemia. To determine the role of AF10$_{PZP}$ in *Hoxa* gene expression, we transduced murine bone marrow-derived HSPCs with either the leukemia-associated CALM-AF10 fusion lacking the first 80 amino acids of AF10, including the first PHD finger (Fig. 1a, second schematic), or a CALM-AF10 fusion (CALM-$_{full}$AF10) which contains full-length AF10 (1–1027 amino acids), including the entire PZP domain, and measured *Hoxa* transcript levels by qRT-PCR. CALM-AF10 expression in murine bone marrow-derived HSPCs led to a substantial increase in *Hoxa7*, *Hoxa9*, *Hoxa10*, and *Meis1* levels compared to the levels of these genes in CALM-$_{full}$AF10 expressing cells, indicating that the exclusion of AF10$_{PZP}$ may be necessary for *HOX/MEIS* activation by the CALM-AF10 fusion protein (Fig. 2d).

To explore whether incorporation of AF10$_{PZP}$ affects *Hoxa* gene activation by CALM-AF10 *in trans*, we transduced CALM-AF10$^{MF}$ leukemia cells with CALM-$_{PZP}$AF10$^{MF}$ and measured *Hoxa* transcript levels by qRT-PCR (Fig. 2e and Suppl. Fig. 1). As

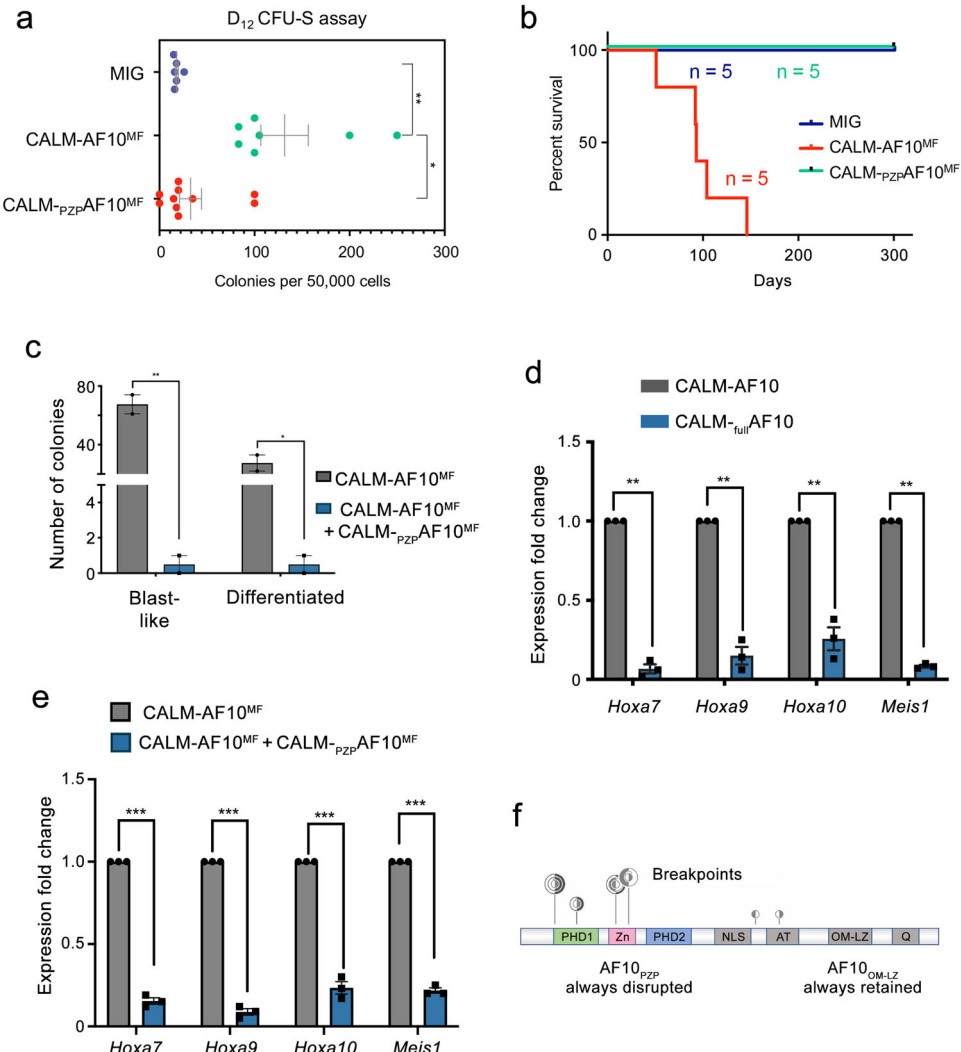

**Fig. 2 AF10_PZP impairs in vivo leukemic activity of CALM-AF10. a** The number of day-12 (D12) colony-forming units in the spleen (CFU-S) enumerated per 50,000 cells from mice injected with each of the indicated constructs are shown. **P value of 0.0015 and *P value of 0.04 (two tailed Student's t-test) and data presented as mean +/− SEM. **b** Kaplan Meier plot showing survival of mice injected with bone marrow-derived HSPCs transduced with MIG (vector), CALM-AF10^MF, or CALM-_PZP_AF10^MF is shown. P value of significance between CALM-AF10^MF and CALM-_PZP_AF10^MF < 0.001 (Log-rank t-test). **c** Number of colonies per 1000 cells with blast-like or differentiated morphology in CALM-AF10^MF leukemia cells or the same cells transduced with the CALM-_PZP_AF10^MF fusion are shown. **P value = 0.009 and *P value =0.03. data are presented as mean +/− SEM from one representative experiment (n = 3). **d** q-RT PCR analysis of expression of Hoxa7, Hoxa9, Hoxa10, and Meis1 in CALM-_full_AF10 transduced murine bone marrow cells shown as fold change relative to expression of these genes in CALM-AF10 expressing cells. ***P value = 0.0003 (two-tailed student's T-test) and data are presented as mean +/− SEM from one representative experiment (n = 3). **e** q-RT PCR analysis of expression of leukemia-associated Hoxa cluster genes and Meis1 in CALM-AF10^MF cells expressing CALM-_PZP_AF10^MF shown as a fold-change relative to expression of these genes in CALM-AF10^MF cells. ***P value = 0.0001 (two-tailed Student's t-test). Data are presented as mean +/− SEM of one representative experiment, (n = 3). **f** Analysis of the TARGET pediatric AML dataset. Translocation breakpoints in the AF10 gene are displayed in the lolliplot. Each vertical line in the lollipop corresponds to an individual fusion breakpoint with the height of the vertical line being proportional to the number of fusions. Different fusions are shown as concentric circles, and the orientation of the filled circle points to the position of AF10 in the fusion, i.e., right fill indicates a 3' AF10 fusion.

expected, CALM-AF10^MF cells were characterized by a high expression of CALM-AF10 target genes Hoxa7, Hoxa9, Hoxa10, and Meis1 (Fig. 2e). A considerable, ~5-10-fold downregulation of these genes observed in the cells transduced with CALM-_PZP_AF10^MF suggested that the CALM-_PZP_AF10^MF fusion can reverse Hoxa activation by the CALM-AF10^MF fusion oncoprotein. Together, our findings demonstrate a key role of AF10_PZP in blocking leukemic transformation by CALM-AF10 through both in cis and in trans mechanisms. These results also help to explain the fact that AF10_PZP is disrupted in all CALM-AF10 fusions, as analysis of the TARGET pediatric AML dataset pointed out that most of the leukemia-associated breakpoints in the AF10 gene in pediatric leukemias are

located in or right after AF10_PZP, and a few more breakpoints are located just upstream of AF10_OMLZ, but importantly, in all these fusions AF10_PZP is impaired or excluded (Fig. 2f).

**AF10_PZP binds to the far N-terminus of histone H3.** The first PHD finger of AF10 (AF10_PHD1) is always impaired in leukemo-genic AF10 fusions (Fig. 2f), and although this may suggest its importance for the normal biological activity of AF10, the function of this domain has not been characterized. Individual PHD fingers are known to recognize H3 tails, either unmodified or methylated at H3K4[18–20], therefore we examined whether AF10_PHD1 has

histone binding activity. We generated [15]N-labeled AF10$_{PHD1}$ and tested it in [1]H,[15]N heteronuclear single quantum coherence (HSQC) NMR experiments. The addition of increasing amounts of the H3$_{1-12}$ peptide (residues 1–12 of H3) to the AF10$_{PHD1}$ sample resulted in large chemical shift perturbations (CSPs) in the AF10$_{PHD1}$ spectrum. CSPs were in the intermediate exchange regime on the NMR timescale and indicated direct and tight interaction (Fig. 3a, left). Titration of the methylated H3K4me3$_{1-12}$ peptide into the AF10$_{PHD1}$ sample led to an overall similar pattern of CSPs, although the magnitude of CSPs induced by H3K4me3 was smaller (Fig. 3a, right). These results suggest that the unmodified H3 peptide and H3K4me3 peptide occupy the same binding site of AF10$_{PHD1}$ and that AF10$_{PHD1}$ slightly prefers an unmodified H3 tail. In agreement, binding of AF10$_{PHD1}$ was ~3-fold tighter to the unmodified H3 peptide (dissociation constant ($K_d$ = 6.5 μM) than to the H3K4me3 peptide ($K_d$ = 22 μM) in physiologically relevant salt concentration of 150 mM, as measured by tryptophan fluorescence (Fig. 3b, c). However, AF10$_{PHD1}$ did not discriminate between unmodified and monomethylated, dimethylated, or trimethylated H3K4 peptides in low, 50 mM salt concentration and bound equally well to all peptides with $K_d$s of ~2–4 μM (Suppl. Fig. 2). No CSPs in AF10$_{PHD1}$ were observed upon titration of the H3$_{3-10}$ peptide (residues 3–10 of H3), implying that AF10$_{PHD1}$ does not bind to H3 lacking Ala1 and Arg2 (Fig. 3d).

Much like AF10$_{PHD1}$, AF10$_{PZP}$ was also capable of binding to the H3 tail, despite the fact that overlay of [1]H,[15]N HSQC spectra of the proteins' apo-states indicated differences in structures (Fig. 3e–h and Suppl. Fig. 3). Comparable $K_d$ values, measured for the interaction of AF10$_{PZP}$ or AF10$_{PHD1}$ with the H3$_{1-12}$ peptide, indicated that the histone binding activity of AF10$_{PHD1}$ is preserved in the context of AF10$_{PZP}$ (Fig. 3e). Peptide pulldown assay further showed that AF10$_{PZP}$ associates with the longer H3$_{1-22}$ and H3$_{1-33}$ peptides and that methylation of H3K4 and H3K9 does not affect this binding, whereas acetylation of lysine residues somewhat reduces it (Fig. 3f–h).

**Structural mechanism of the AF10$_{PZP}$-H3$_{1-12}$ interaction.** To define the molecular basis for the interaction of AF10$_{PZP}$ with the histone H3 tail, we generated a fusion construct that contains residues 1–12 of H3 covalently linked to the residues 19-208 of AF10 via a short GSGSS linker. We note that the position of the H3 sequence in the linked construct, which is N-terminal to the sequence of AF10$_{PZP}$, was critical because a free Ala1 of H3 is required for the interaction with AF10$_{PHD1}$ (Fig. 3d). The [1]H,[15]N HSQC spectrum of the [15]N-labeled linked H3$_{1-12}$-AF10$_{PZP}$ construct overlaid well with the [1]H,[15]N HSQC spectrum of isolated AF10$_{PZP}$ recorded in the presence of a five-fold excess of the H3$_{1-12}$ peptide, indicating that the linked and unlinked complexes adopt similar structures in solution (Suppl. Fig. 4). The fusion protein was crystallized, and the structure of the H3-bound AF10$_{PZP}$ was determined to a 2.1 Å resolution (Fig. 4 and Suppl. Table 1).

The structure revealed a saddle-like globular fold of AF10$_{PZP}$ comprised of five zinc-binding clusters (Fig. 4a, b). The Ala1-Thr6 residues of the H3 tail occupied an extended groove of AF10$_{PHD1}$ with Arg2-Lys4 forming an anti-parallel β strand that paired with the protein's β1-β2 sheet, whereas Ala7-Gly12 residues of H3 curved away from the protein surface. Characteristic β-sheet interactions were observed between the backbone amides of Arg2 and Lys4 of H3 and Y41 and L39 of AF10$_{PZP}$. The N-terminal amino group of Ala1 of H3 was engaged through hydrogen bonds with the backbone carbonyl groups of P62, T63, and G64 of the protein (Fig. 4b, c). The guanidino group of Arg2 donated hydrogen bonds to the side-chain carboxyl group of D43 and the backbone carbonyl of C42. The side chain amino moiety of Lys4 was restrained through a hydrogen bond with the

backbone carbonyl of E31. The side-chain amide of Gln5 formed a hydrogen bond with the backbone carbonyl of A35, whereas the backbone carbonyl of Thr6 was hydrogen-bonded to the backbone amide of G33. Overall, the structural mode of the AF10$_{PZP}$-H3$_{1-12}$ interaction was reminiscent that of observed for the PZP domain of BRPF1[21,22].

**AF10$_{PZP}$ recognizes two regions of the H3 tail.** AF10$_{PZP}$ has previously been shown to associate with a region of H3 spanning residues 21–27 but not to bind H3$_{1-21}$ peptide[15]. While the presented here structure of H3$_{1-12}$-AF10$_{PZP}$ clearly demonstrates the interaction between AF10$_{PZP}$ and the far N-terminal part of H3, particularly residues Ala1-Thr6, in the previously reported structure of the AF10$_{PZP}$-H3$_{1-36}$ fusion, AF10$_{PZP}$ associates with the middle part of H3 (Ala21-Lys27)[15]. An overlay of these structures shows that the two regions of the H3 tail occupy different binding sites of AF10$_{PZP}$ (Fig. 5a). While the N-terminal region of H3 (yellow) is bound by AF10$_{PHD1}$, the middle region of H3 (magenta) is bound at the interface of the PHD fingers and the zinc knuckle.

To gain insight into the binding of AF10$_{PZP}$ to the H3 tail, we performed [1]H,[15]N HSQC NMR titration experiments using H3 peptides of different sizes (Fig. 5b–d and Suppl. Figs. 5–7). Titration of either H3$_{1-12}$ peptide or H3$_{15-34}$ peptide to the AF10$_{PZP}$ NMR sample led to dissimilar patterns of CSPs, confirming that the two peptides are bound in separate binding pockets of AF10$_{PZP}$ (Fig. 5b, c). In both titrations, CSPs were in the intermediate exchange regime, which was in agreement with $K_d$s of 7.5 μM and 2.2 μM measured for the interaction of AF10$_{PZP}$ with H3$_{1-12}$ peptide and H3$_{15-34}$ peptide, respectively (Fig. 5e, f). The longer H3 peptide (H3$_{1-31}$), however, was bound tighter by AF10$_{PZP}$. Analysis of the fluorescence-derived binding curves for the AF10$_{PZP}$-H3$_{1-31}$ interaction required a two-site binding model, and the fitting yielded two $K_d$ values of 0.3 μM and 5.9 μM, suggesting a cooperative engagement of the two regions of H3$_{1-31}$ (Fig. 5e, g). In support, CSPs in a slow exchange regime, indicative of a tight interaction, were observed in the AF10$_{PZP}$ NMR spectra upon titration with the H3$_{1-31}$ peptide (Fig. 5d).

**An almost entire H3 tail is engaged with AF10$_{PZP}$.** Analyzing the crystal structures of the H3$_{1-12}$-AF10$_{PZP}$ and AF10$_{PZP}$-H3$_{1-36}$ complexes (Fig. 5a), we generated AF10$_{PZP}$ mutants which are impaired in binding to either the Ala1-Thr6 region of H3 or the Ala21-Lys27 region of H3. Particularly, the AF10$_{PZP}$ E179K mutant lost its ability to bind to the H3$_{15-34}$ peptide in NMR titration experiments but retained the ability to bind to H3$_{1-12}$ and H3$_{1-31}$ peptides through the interaction with the far N-terminal part of H3 (Fig. 6a, b and Suppl. Figs. 8 and 9). Binding affinities of AF10$_{PZP}$ E179K for the H3$_{1-12}$ and H3$_{1-31}$ peptides (8.5 μM and 7.8 μM) were essentially the same as the binding affinity of WT AF10$_{PZP}$ for the H3$_{1-12}$ peptide (7.5 μM) (Figs. 6c–e and 5e). Conversely, the AF10$_{PZP}$ D43K mutant was defective in binding to the H3$_{1-12}$ peptide but retained the ability to bind to H3$_{15-34}$ and H3$_{1-31}$ peptides through the interaction with the middle part of H3 (Figs. 6e–h and 5e and Suppl. Fig. 10). Binding affinities of AF10$_{PZP}$ D43K for the H3$_{15-34}$ and H3$_{1-31}$ peptides (2.2 μM and 2.9 μM) were similar to the binding affinity of WT AF10$_{PZP}$ for the H3$_{15-34}$ peptide (Figs. 6e and 5e). Pulldown assays using biotinylated histone peptides and the GST-AF10$_{PZP}$ mutants supported the conclusion derived from the NMR data and measurements of binding affinities: disruption of either binding pocket of AF10$_{PZP}$, although decreases, does not abolish binding to H3. The double D43K/E179K mutation in both

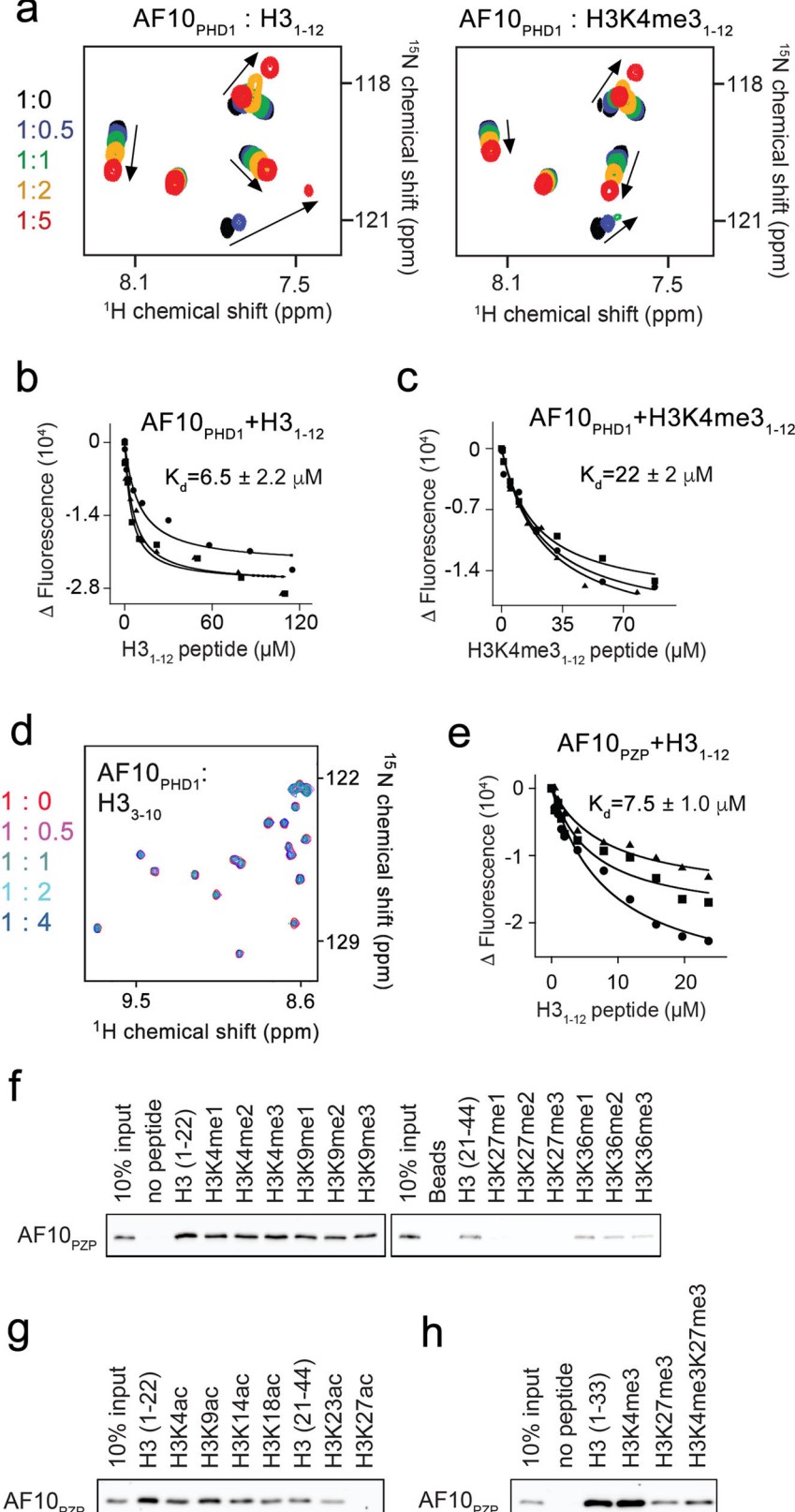

**Fig. 3 AF10$_{PZP}$ binds to the N-terminus of the H3 tail. a** Overlay of $^1$H,$^{15}$N HSQC spectra of AF10$_{PHD1}$ in the presence of the increasing amount of H3$_{1-12}$ or H3K4me3$_{1-12}$ peptide. Spectra are colored according to the protein:peptide molar ratio. **b, c** Binding curves used to determine $K_d$ values by tryptophan fluorescence. $K_d$s are represented as mean values +/− S.D. from three independent experiments ($n = 3$). **d** Overlay of $^1$H,$^{15}$N HSQC spectra of AF10$_{PHD1}$ in the presence of the increasing amount of H3$_{3-10}$ peptide. Spectra are colored according to the protein:peptide molar ratio. **e** Binding curves used to determine $K_d$ by tryptophan fluorescence. $K_d$ is represented as mean +/− S.D. from three independent experiments ($n = 3$). **f–h** Histone peptide pulldown assays of GST-AF10$_{PZP}$ with the indicated biotinylated peptides.

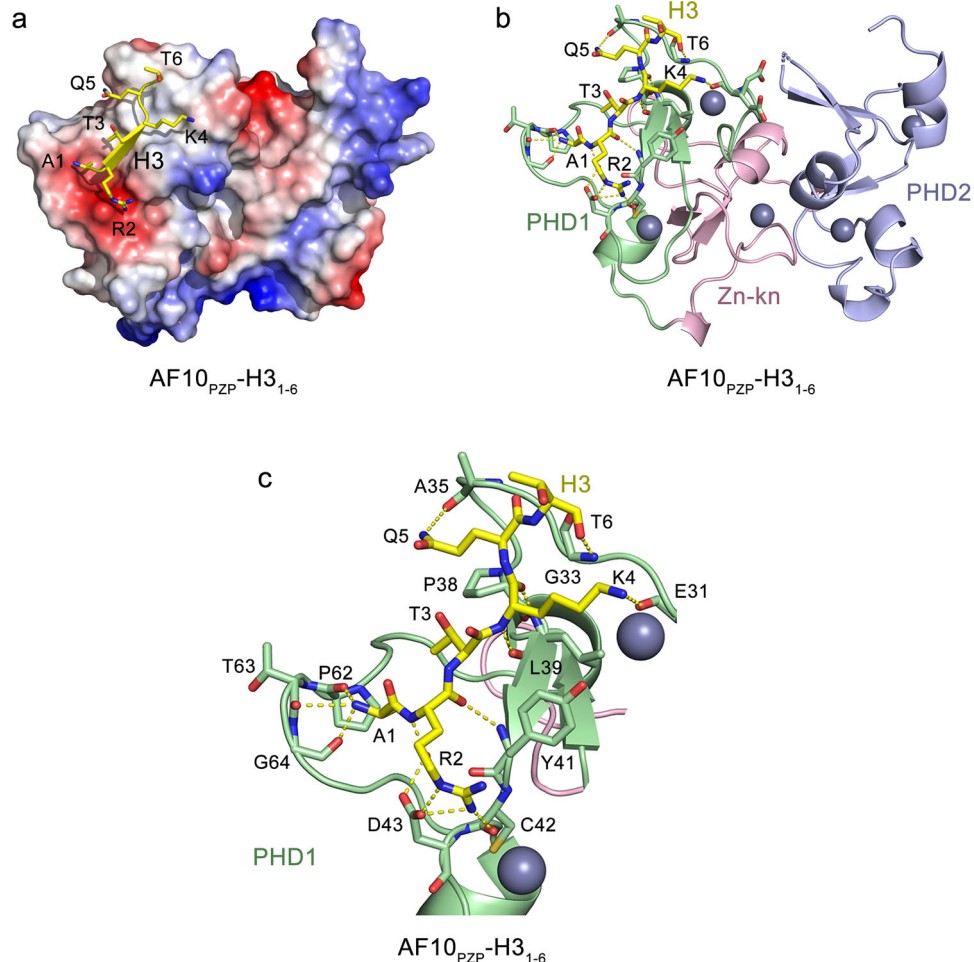

**Fig. 4 Structural basis for the recognition of H31-6 by AF10PZP. a** Electrostatic surface potential of AF10PZP in the complex, with blue and red colors representing positive and negative charges, respectively. The H3 tail (only Ala1-Thr6 of H3 are shown for clarity) is yellow. **b** The crystal structure of the H3-bound AF10PZP is shown as a ribbon diagram with PHD1, Zn-knuckle, and PHD2 colored light green, pink, and light blue, respectively. The Ala1-Thr6 region of H3 is depicted as yellow sticks. The zinc ions and waters are shown as gray and red spheres, respectively. Yellow dashed lines indicate hydrogen bonds. **c** Close-up view of the histone H31-6 binding site of AF10PZP.

sites of AF10PZP is required to eliminate the interaction with H3 (Fig. 6i, j).

Can AF10PZP engage both the far N-terminal region and the middle region of H3 simultaneously? We addressed this question via a reverse NMR titration experiment. We produced $^{15}$N-labeled H3 tail (residues 1–44) and recorded its $^1$H,$^{15}$N HSQC spectra while adding unlabeled AF10PZP to the sample (Fig. 6k). Synergetic resonance changes, including cross peak disappearance and shifts, were detected in all observable backbone amides between Gln5 and Ala29 of H3, suggesting that the entire Ala1-Lys27 region of the H3 tail was perturbed and therefore likely involved in the interaction. A model of the H3$_{1-31}$-AF10PZP complex generated using the simulated annealing method and both crystal structures revealed that the two regions can be bound by AF10PZP simultaneously *in cis* (Fig. 6l).

**AF10PZP associates with both H3 and DNA within the nucleosome.** To explore the histone binding mechanism of AF10PZP in the context of chromatin, we tested the interaction of AF10PZP with the nucleosome core particle (NCP) in electrophoretic mobility shift assays (EMSA) and fluorescence anisotropy assays (Fig. 7a–e). We reconstituted NCP using a 207 bp DNA (NCP$_{207}$) in which 147 bp 601 DNA is flanked by 30 bp linker DNA on either side and internally labeled with fluorescein

27 bp in from the 5' end. NCP$_{207}$ was incubated with increasing amounts of AF10PZP, WT, and mutants, and the reaction mixtures were resolved on a 5% native polyacrylamide gel (Fig. 7a–c). A gradual increase in the amount of added WT AF10PZP resulted in a shift of the NCP$_{207}$ band, indicative of the formation of the AF10PZP-NCP$_{207}$ complex, but this shift was delayed when either AF10PZP D43K mutant or E179K mutant were used, implying that interaction of AF10PZP with H3 tail is important for binding to the nucleosome. However quantitative measurement of binding affinities by fluorescence polarization revealed that the decrease in binding to NCP$_{207}$ due to D43K or E179K mutation was modest. Titration of WT AF10PZP against NCP$_{207}$ yielded an $S_{1/2}$ of 6 µM for the AF10PZP-NCP$_{207}$ complex formation, whereas binding of the D43K and E179K mutants was only slightly weaker ($S_{1/2} = 9$ µM and 14 µM, respectively) (Figs. 7d and 6e). The association of WT AF10PZP with the nucleosome reconstituted with 147 bp 601 DNA (NCP$_{147}$) was also reduced ($S_{1/2} = 15$ µM), suggesting that the extra-nucleosomal linker DNA contributes to the interaction of AF10PZP with NCP$_{207}$ (Figs. 7e and 6e). This observation prompted us to investigate whether AF10PZP can also bind DNA. Indeed, a decrease in band intensity of 147 bp 601 DNA upon addition of GST-AF10PZP in EMSA and CSPs induced in AF10PZP by 147 bp 601 DNA in $^1$H,$^{15}$N HSQC experiments indicated that AF10PZP binds to DNA

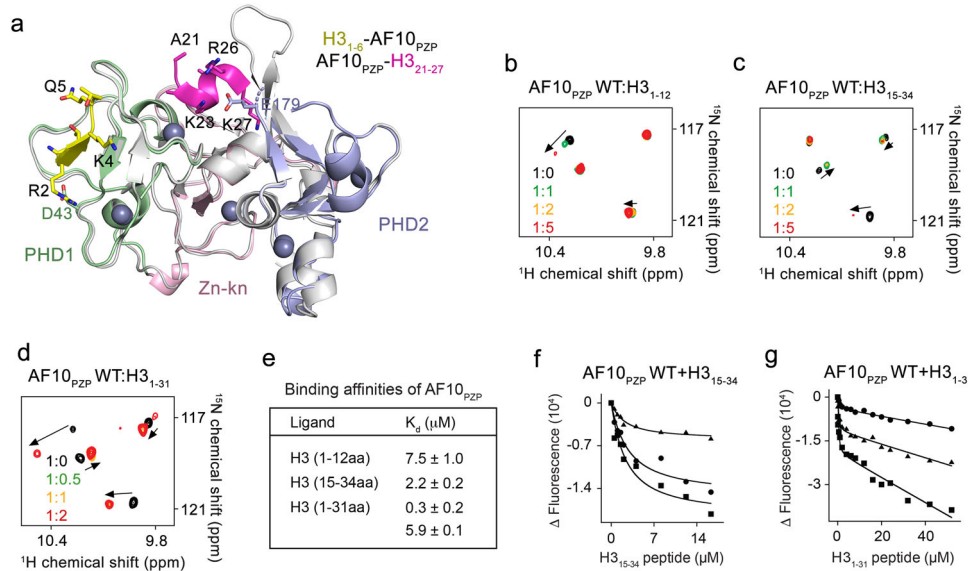

**Fig. 5 AF10$_{PZP}$ can interact with two regions of the H3 tail. a** Overlay of the structures of the complexes H3$_{1-6}$-AF10$_{PZP}$, colored as in Fig. 4b, and AF10$_{PZP}$-H3$_{21-27}$ (PDB ID 5DAH). The histone regions H3$_{1-6}$ and H3$_{21-27}$ are yellow and magenta, respectively. D43 and E179 residues mutated in this study are also shown. **b–d** Superimposed $^1$H,$^{15}$N HSQC spectra of AF10$_{PZP}$ collected upon titration with indicated H3 peptides. Spectra are color-coded according to the protein:peptide molar ratio. **e–g** Binding affinities of AF10$_{PZP}$ for the indicated histone peptides as measured by tryptophan fluorescence. $K_d$s are represented as mean values +/− S.D. from three independent experiments ($n = 3$). **f, g** Binding curves used to determine $K_d$s in **e**.

(Fig. 7f, g and Suppl. Fig. 11). These results were further substantiated by fluorescence anisotropy assays, which yielded an S$_{1/2}$ of 8 µM for the interaction of AF10$_{PZP}$ with fluorescently labeled 207 bp 601 DNA (Fig. 7h).

Collectively, our structural and biochemical studies suggest a model for the AF10$_{PZP}$ engagement with the nucleosome, a fundamental unit of chromatin. AF10$_{PZP}$ binds to almost the entire H3 tail, wrapping the tail around and also associates with DNA. Methylation of H3K4 largely does not affect histone binding activity of AF10$_{PZP}$ (Figs. 3f, h and 6i, j), however, acetylation of H3K23 (Suppl. Fig. 12) or methylation or acetylation of H3K27 (Figs. 3f–h and 6i, j) considerably decrease this interaction, in agreement with the previous studies[15]. The binding of AF10$_{PZP}$ to NCP does not alter the nucleosome dynamics, because no measurable changes were detected in Cy3-Cy5 labeled NCP$_{147}$ in FRET assays (Suppl. Fig. 13).

**AF10$_{PZP}$ promotes nuclear localization of CALM-AF10$^{MF}$ and is required for association with chromatin**. To examine the role of AF10$_{PZP}$ in the sub-cellular localization of CALM-AF10, we transfected Flag-tagged CALM-AF10$^{MF}$ and CALM-$_{PZP}$AF10$^{MF}$ into HEK 293T cells and visualized the proteins by immunofluorescence (IF) using an anti-Flag antibody. IF analysis showed that while the CALM-AF10$^{MF}$ fusion protein was predominantly cytosolic, in support of previous findings[23,24], the CALM-$_{PZP}$AF10$^{MF}$ fusion protein accumulated largely in the nucleus (Fig. 8a). Such a shift in the sub-cellular distribution pointed to a crucial role of AF10$_{PZP}$ in promoting the nuclear localization of CALM-$_{PZP}$AF10$^{MF}$. Furthermore, CALM-$_{PZPmut}$AF10$^{MF}$ fusion protein, harboring D43K/E179K mutations that disrupt binding to H3 tail, lost its ability to accumulate in the nucleus and was found primarily in the cytosol, confirming the importance of functional AF10$_{PZP}$ for the nuclear pool of CALM-AF10 (Fig. 8a, right panels).

To assess the ability of AF10$_{PZP}$ to bind chromatin, we investigated the genomic occupancy of AF10$_{PZP}$ in the MOLM13 human leukemia cell line. We cloned AF10$_{PZP}$ with 2× nuclear localization signals and stably transduced MOLM13 cells. Chromatin immunoprecipitation followed by sequencing

(ChIP-seq) using a custom-made antibody directed against AF10$_{PZP}$ showed that AF10$_{PZP}$ co-localizes with the transcription start sites of numerous genes (Fig. 8b). In agreement with in vitro binding data, in cells AF10$_{PZP}$ occupied chromatin regions enriched in H3K4me3 (as well as H3K79me2), however did not bind to the chromatin sites enriched in H3K27me3. The inhibition of chromatin binding activity of AF10$_{PZP}$ by the repressive H3K27me3 methylation mark appears to be very strong as no enrichment of AF10$_{PZP}$ was observed at bivalent promoters associated with both H3K4me3 and H3K27me3 marks (Fig. 8c, d), which is also consistent with histone peptide pulldown results (Fig. 6i, j).

**AF10$_{PZP}$ increases the spreading of H3K79me2**. The CALM-AF10 fusion is believed to play a role in targeting DOT1L to gene loci, which results in the deposition of H3K79 methylation and transcriptional activation. We, therefore, examined whether the inclusion of AF10$_{PZP}$ in CALM-AF10 can lead to changes in H3K79 methylation in CALM-AF10 leukemia cells. We performed ChIP-seq experiments using in trans leukemia repression system, in which CALM-$_{PZP}$AF10$^{MF}$ was overexpressed in CALM-AF10$^{MF}$ leukemia cells (Fig. 2e). ChIP-seq analysis showed that incorporation of AF10$_{PZP}$ by overexpressing CALM-$_{PZP}$AF10$^{MF}$ led to the gain of H3K79me2 at a number of new genomic sites (Fig. 9a). In contrast, there were almost no sites associated with the loss of H3K79me2 upon CALM-$_{PZP}$AF10$^{MF}$ overexpression. Furthermore, the incorporation of AF10$_{PZP}$ caused the spreading of H3K79me2 genome-wide beyond the H3K79me2-enriched sites in CALM-AF10$^{MF}$ leukemia cells (Fig. 9b). We note that most of the increase in H3K79me2 levels was found at promoter-proximal regions of genes, including those that are not CALM-AF10 targets (Figs. 9b and 10a). These results suggest that similar to overexpression of DOT1L or AF10 in leukemia cells[25], the inclusion of AF10$_{PZP}$ leads to H3K79me2 spreading and reversal of leukemogenesis.

In conclusion, our findings indicate that genomic rearrangements of AF10 in leukemia disrupt the intricate relationship between chromatin binding function of AF10$_{PZP}$ and chromatin

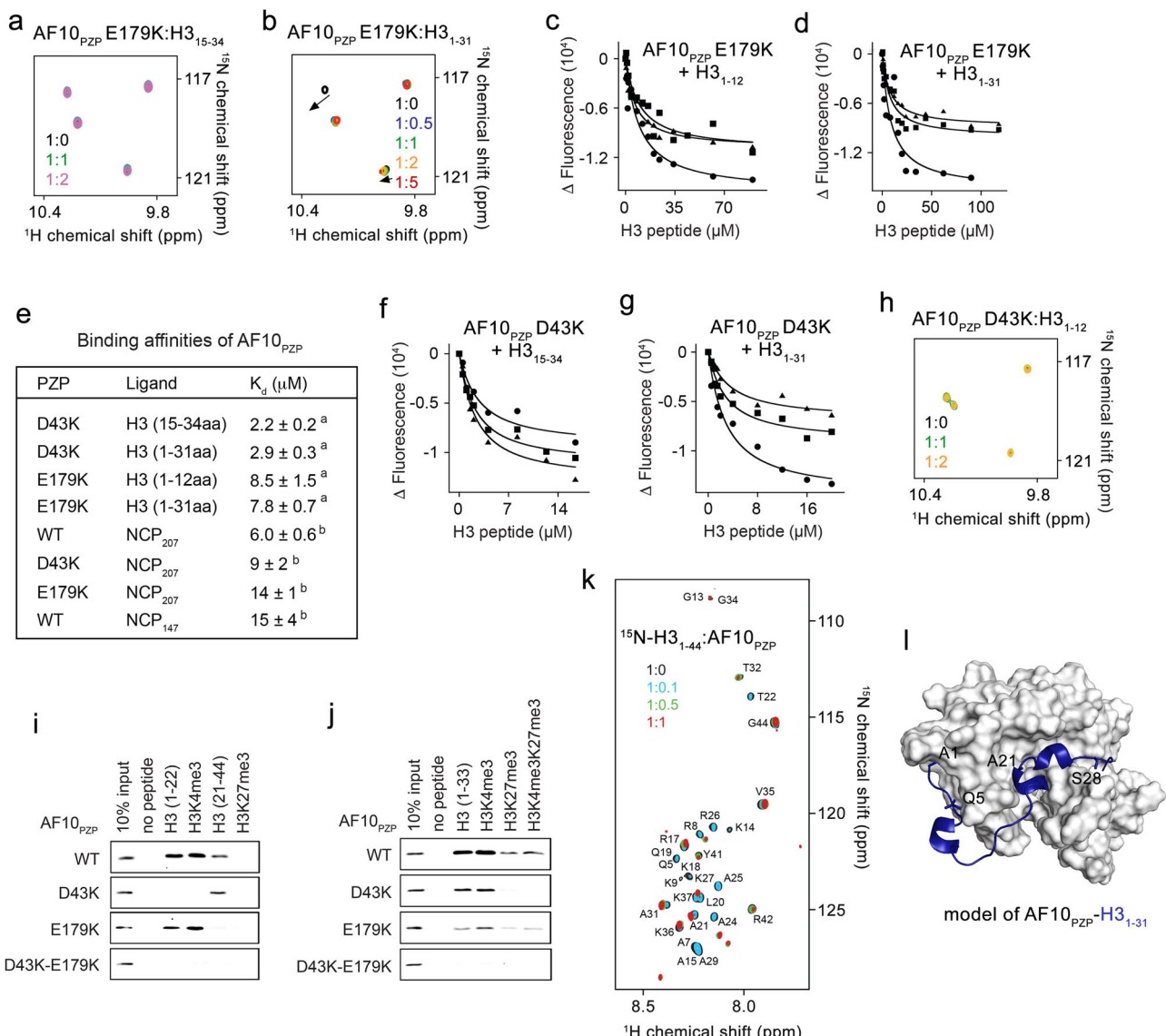

**Fig. 6 An almost entire H3 tail is engaged with AF10_PZP. a, b** Superimposed $^1$H,$^{15}$N HSQC spectra of the AF10_PZP E179K mutant collected upon titration with indicated H3 peptides. Spectra are color-coded according to the protein:peptide molar ratio. (**c, d**) Binding curves used to determine $K_d$s by tryptophan fluorescence. **e** Binding affinities of WT and mutated AF10_PZP for the indicated ligands as measured by ($^a$) tryptophan fluorescence and ($^b$) fluorescence anisotropy. $K_d$s are represented as mean values +/− S.D. from three independent experiments ($n = 3$). **f, g** Binding curves used to determine $K_d$s by tryptophan fluorescence. **h** Superimposed $^1$H,$^{15}$N HSQC spectra of the AF10_PZP D43K mutant collected upon titration with H3$_{1-12}$ peptide. Spectra are colored according to the protein:peptide molar ratio. **i, j** Histone peptide pulldown assays of WT and mutated GST-AF10_PZP with the indicated biotinylated peptides. **k** Superimposed $^1$H,$^{15}$N HSQC spectra of the histone H3$_{1-44}$ tail collected upon titration with unlabeled AF10_PZP. Spectra are color-coded according to the histone:AF10_PZP molar ratio. **l** A model for the association of AF10_PZP with the histone H3$_{1-31}$ tail (blue) generated using Xplor 2.14.

methylation by DOT1L, leading to the establishment and/or perpetuation of oncogenic transcriptional programs. This view is supported by the observation that AF10 fusions invariably exclude the chromatin reader–AF10_PZP in leukemia while always retaining AF10_OMLZ and thus enabling DOT1L-mediated histone H3K79 methylation. We show that AF10_PZP engages the nucleosome through multivalent contacts with histone H3 tail and DNA and binds to chromatin in cells, colocalizing with active methylation marks and discriminating against the repressive H3K27me3 mark. Our results demonstrate that CALM_PZPAF10$^{MF}$ decreases *Hoxa* gene expression in CALM-AF10$^{MF}$ leukemia cells and that incorporation of AF10_PZP in the leukemogenic fusion blocks the transforming activity in vitro and in vivo and abolishes CALM-AF10-driven leukemogenesis in vivo.

Altogether, our data suggest the molecular mechanism underlying the leukemogenic activity of the CALM-AF10 fusion (Fig. 10b). It has been shown that the nuclear export receptor CRM1 recruits CALM-AF10 to *Hoxa* loci via binding to the nuclear export signal of CALM[26]. In the absence of functional AF10_PZP within the leukemogenic fusion, CALM-AF10 can trap DOT1L at the *Hoxa* cluster, leading to the elevated local H3K79me2 level, constitutive activation of *Hoxa* genes, and a decrease in global H3K79me2 level due to the inability of the fusion to spread onto chromatin regions beyond the *Hoxa* loci (Fig. 10b, top). Incorporation of the chromatin reader, AF10_PZP in the CALM-AF10 fusion allows for spreading onto other regions of chromatin, thus disseminating DOT1L to other sites in the genome. This mechanism sheds light on the aberrant stabilization

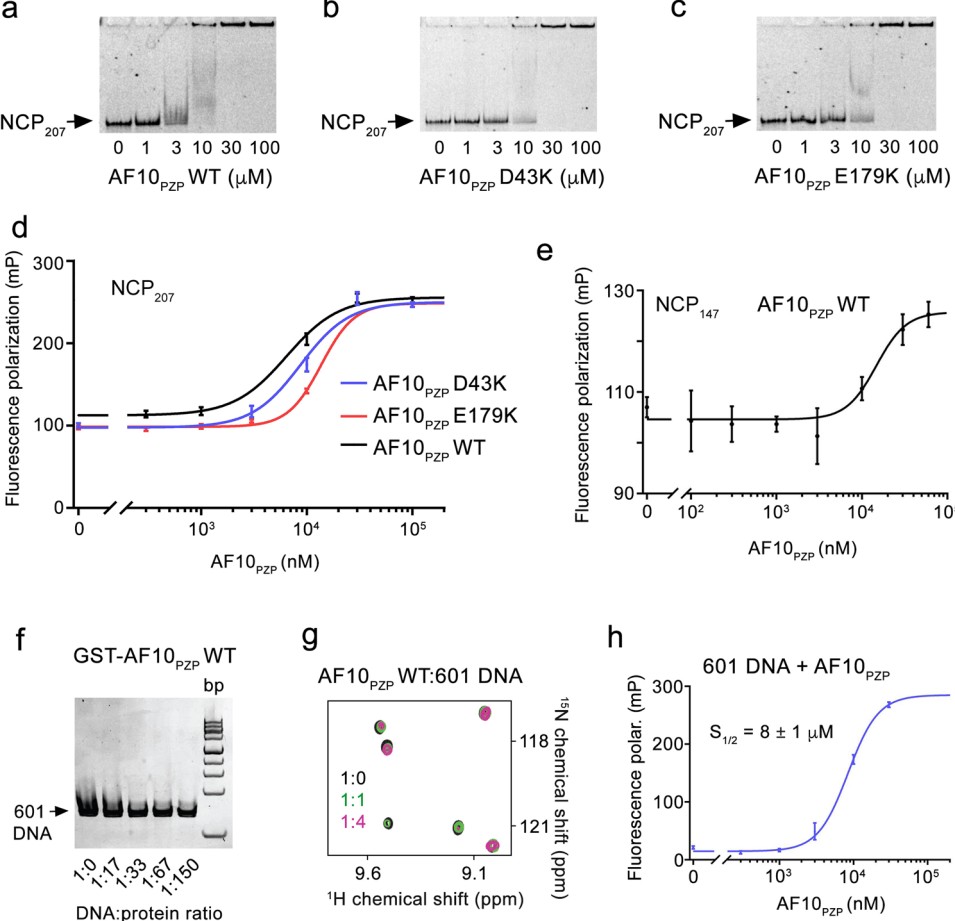

**Fig. 7 AF10$_{PZP}$ binds to the H3 tail and DNA within the nucleosome. a–c** EMSA with NCP$_{207}$ in the presence of increasing amounts of WT and mutated AF10$_{PZP}$. The amount of AF10$_{PZP}$ mixed with 5 nM NCP$_{207}$ is shown below gel images. **d, e** Binding curves for the interactions of WT and mutated AF10$_{PZP}$ with NCP$_{207}$ (**d**) and NCP$_{147}$ (**e**) as measured by fluorescence polarization. Data are represented as mean values $+/-$ S.D. from three independent experiments ($n = 3$). Binding affinities are summarized in Fig. 6e. **f** EMSA with 147 bp 601 DNA in the presence of the increasing amount of WT GST-AF10$_{PZP}$. DNA to protein molar ratio is shown below the gel image. **g** Overlay of $^1$H,$^{15}$N HSQC spectra of AF10$_{PZP}$ in the presence of the increasing amount of 147 bp 601 DNA. Spectra are colored according to the protein:DNA molar ratio. **h** The binding affinity of AF10$_{PZP}$ for 207 bp 601 DNA was measured by fluorescence polarization. S$_{1/2}$ is represented as mean $+/-$ S.D. from three independent experiments ($n = 3$).

of DOT1L at critical oncogenes and points to the CALM-AF10 fusion as a potential candidate for gene therapy aiming to eliminate the upregulation of oncogenes and reverse leukemogenesis.

## Methods

**Plasmids and constructs**. The p-MIG-CALM-AF10 and pMIY-CALM-AF10$^{MF}$ constructs have been described previously[17]. For the CALM-$_{full}$AF10, a PCR amplified full-length AF10 fragment (corresponding to amino acids 1-1027) was cloned downstream of the CALM part of the pMIG-CALM-AF10 construct, also amplified by PCR. For the CALM-$_{PZP}$AF10$^{MF}$ construct, a PCR amplified fragment corresponding to amino acids 1–197 of AF10 (ENST00000377072.8) was PCR amplified and cloned into the CALM-AF10$^{MF}$ fusion construct using the BamH1 site in between the CALM and AF10 portions. Primers used in this study are listed in the source data file.

**Mice and bone marrow transduction**. Parental strain mice were bred and maintained at the Helmholtz Centre Munich, Animal Resources at Children's Hospital (ARCH), or the SBP animal facility. All animal experiments described in this study were approved by and adhered to the guidelines of the Sanford Burnham Prebys, Children's Hospital Boston, or Helmholtz Center Institutional Animal Care and Use Committees under approved protocols. Lineage −ve (lin depleted) cells from murine bone marrow were isolated either by using Mouse hematopoietic progenitor cell isolation kit (STEMCELL Technologies, Canada) as per the manufacturer's protocol or by injecting donor mice with 5-FU. Five days post 5-FU injection, bone marrow from these mice were harvested by crushing of femur and tibia and plated in bone marrow medium (Dulbecco's modified Eagle's medium, 15% fetal bovine serum, 1% Pen/Strep) + cytokines (100 ng/ml stem cell factor,

10 ng/ml interleukin 6 (IL6), 6 ng/ml interleukin 3 (IL3)). Forty-eight hours after prestimulation of the bone marrow cells, they were transduced with different viruses by overlaying them on virus-producing irradiated (400 cGy) GP+E86 producers in the presence of cytokines and protamine sulfate (5 μg/mL) or by spinfection with virus conditioned medium (VCM). These cells were then sorted for GFP or YFP expression using a FACSVantage (Becton Dickinson, Franklin Lakes, NJ, USA) or BD FACSAria II (BD Biosciences, US) flow sorting machine. Sorted GFP or YFP-positive cells were used for colony-forming cell (CFC) or colony-forming unit-spleen (CFU-S) assays or qRT-PCR or injected directly into recipient mice.

**Bone marrow isolation and murine transplantation assays**. CALM-AF10$^{MF}$ leukemia cells were transduced with the MIG empty vector or the MIG-CALM-$_{PZP}$AF10$^{MF}$ vector-expressing viruses and sorted for GFP/YFP expression. Following sorting, 200,000 leukemia cells from these two arms were injected into 800 cGy irradiated C57BL/6J mice through tail vein injections. Hematopoietic engraftment of GFP or YFP-positive cells was assessed by flow cytometry of regularly collected peripheral blood samples. Mice were closely monitored for signs of disease manifestation and sacrificed when they showed signs of leukemic disease.

**Colony-forming unit assays**. For CFU assays, GFP or YFP sorted cells were counted and plated in 1% myeloid-conditioned methylcellulose containing Iscove's modified Dulbecco medium-based Methocult (Methocult M3434; StemCell Technologies, Vancouver, Canada) at a concentration of 1000 cells/mL.

**CFU-S assays**. Bone marrow cells from 5-fluorouracil-treated mice were isolated, transduced with retroviral supernatants from various constructs, sorted and

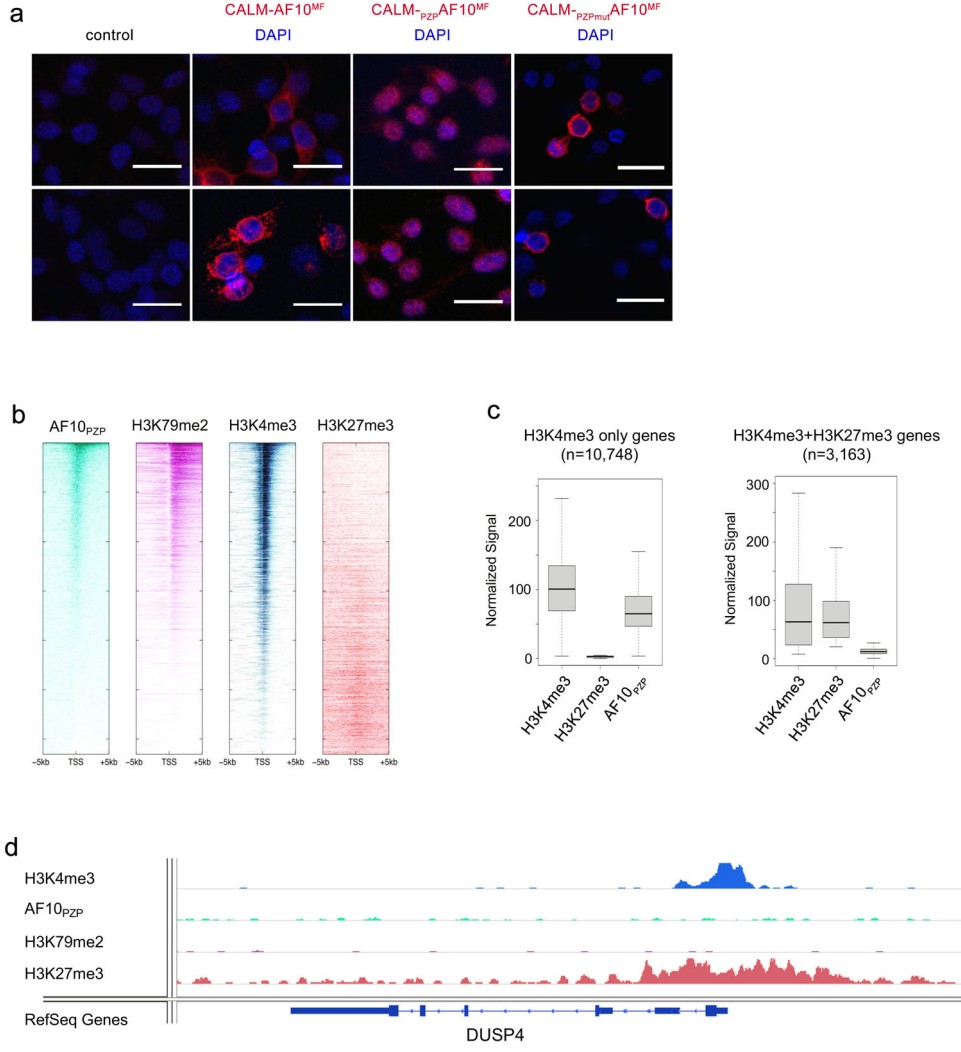

**Fig. 8 AF10$_{PZP}$ promotes nuclear localization of CALM-AF10$^{MF}$ and is required for association with chromatin. a** Immunofluorescence image of 293T cells transfected with 1XFLAG CALM-AF10$^{MF}$, 1XFLAG CALM-$_{PZP}$AF10$^{MF}$, or 1XFLAG CALM-$_{PZPmut}$AF10$^{MF}$ and probed with anti-FLAG antibody (red). DAPI (blue) was used for nuclear staining. Non-transfected cells were used as control. Scale bar, 20 μm. ($n = 3$) **b** Heatmaps showing AF10$_{PZP}$ peaks centered around transcription start sites (TSS) in MOLM13 cells, as well as H3K79me2, H3K4me3, and H3K27me3 at the same loci sorted by decreasing AF10$_{PZP}$ binding. **c** Normalized ChIP-seq signals at the genes marked with H3K4me3 (10,748 genes) and with both H3K4me3 and H3K27me3 (3163 bivalent genes). **d** The genomic locus of a representative H3K4me3/H3K27me3 bivalent gene, DUSP4 is shown.

injected intravenously into lethally irradiated (800 cGy of 137Cs γ-radiation) (C57BL/6J × C3H/HeJ) F$_1$ (B6C3) mice at cell numbers adjusted to give 5 to 15 macroscopic spleen colonies. The number of macroscopic colonies was visualized after sacrificing the mice 12 days after injection, fixing the spleen in Telleyesniczky solution (absolute ethanol, glacial acetic acid, and formaldehyde mixed in a 9:1:1 ratio, respectively). For the CALM-AF10$^{MF}$ mutant, mice were injected with fewer cells to ensure scoring resolution (1000 GFP sorted cells per mouse).

**ChIP and ChIP-seq.** For AF10$_{PZP}$ ChIP-seq, MOLM13 cells stably transduced with the retrovirally delivered AF10$_{PZP}$ construct were used for chromatin immuno-precipitation (ChIP) with a custom antibody generated against AF10$_{PZP}$. Immu-noprecipitation was performed as described earlier[13]. Thirty million cells were fixed using 1% formaldehyde and chromatin was sheared using Diagenode Bioruptor for 15 min with 15 cycles (each 30 s on, 30 s off-cycle) setting at 4 °C.

ChIP-seq for H3K79me2 was performed on 1 million CALM-AF10$^{MF}$ leukemia cells or the same cells transduced with the pMIG-CALM-$_{PZP}$AF10$^{MF}$ virus and sorted for GFP 72 h after transduction and used directly for fixing and sonication as described above. The amount of each antibody used for ChIP experiments is listed in the source data file. Library preparation on eluted DNA was performed using the NEBNext Ultra II DNA library prep kit for Illumina (E7645S and E7600S) as per the manufacturer's protocol. Library prepped DNA was subjected to sequencing by NextSeq 500 (Illumina, La Jolla, CA) at the Genomics core, MSKCC (New York, NY).

**RNA isolation and qRT-PCR.** RNA was extracted using RNeasy Mini kit (Qiagen) according to the manufacturer's recommendations and cDNA was prepared using oligo(dT) primers and the SuperScript® III First-Strand Synthesis System (Thermo Fisher, Carlsbad, CA). cDNA was quantified by NanoDrop and used for q-RT-PCR assays with Taqman probes for *Hoxa* genes, *Meis1* and *Gapdh* or *B-Actin* genes. Taqman probe information will be provided on request. q-RT-PCR was performed on the ABI 96-well PCR system, and data were analyzed by the delta-delta Ct method.

**Immunofluorescence.** 293T cells were seeded on coverslips and transfected with 1XFLAG CALM-AF10$^{MF}$, 1XFLAG CALM-$_{PZP}$AF10$^{MF}$, or 1XFLAG CALM-$_{PZPmut}$AF10$^{MF}$. Non-transfected cells were used as controls. After 48 h of transfection, cells were washed with 1× PBS once and fixed with 2% paraf-ormaldehyde/PBS solution for 10 mins. Cells were air-dried briefly for 2–3 mins, then washed with 1× PBS for 3 mins and permeabilized in 0.1% Triton X for exactly 5 mins. After washing with 1× PBS, cells were blocked in PBS containing 3% BSA + 0.1% Tween 20. Cells were incubated in anti-FLAG M2 (Sigma F1804, 1:500, 2 μg/mL) primary antibody in blocking buffer at 4 °C overnight. The fol-lowing day, cells were washed 3 times with PBS + 0.1% Tween 20 for 5 mins each and then incubated with Alexa Fluor 647 goat anti-mouse secondary antibody (Molecular Probes A-21236, 1:1000, 2 μg/mL) in blocking buffer for 1 h at room temperature in dark/protected from light. Cells were then washed and mounted onto glass slides in ProLong Diamond Antifade Mountant with DAPI (Molecular Probes). Images were acquired with Zeiss LSM 710 NLO confocal microscope at ×40 objective.

a

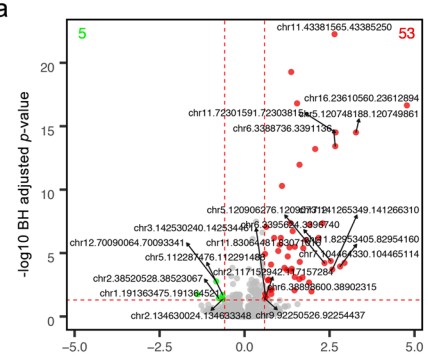

b

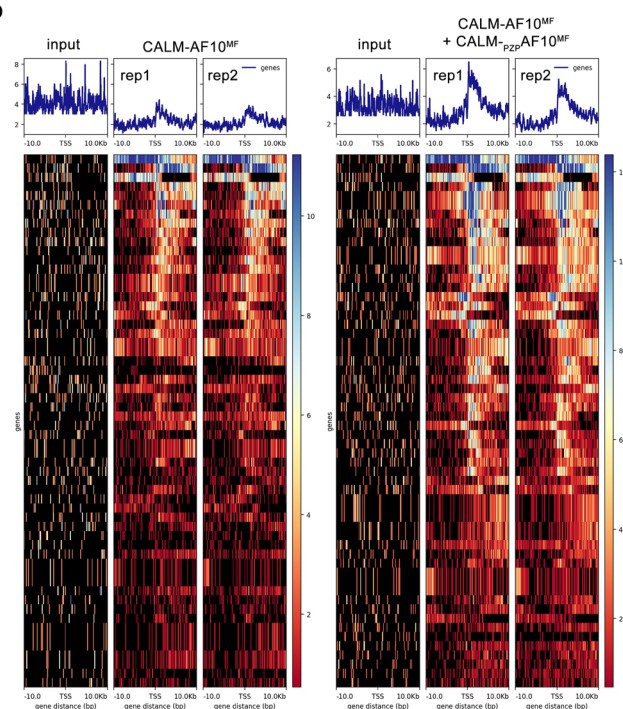

**Fig. 9 AF10$_{PZP}$ increases the spreading of H3K79me2. a** Volcano plot showing genomic regions with differential H3K79me2 distribution in CALM-AF10$^{MF}$ leukemia cells and in the same cells expressing CALM-$_{PZP}$AF10$^{MF}$. Red dots represent regions that gained H3K79me2, green dots represent regions that lost H3K79me2 peaks, and gray dots represent the regions with statistically insignificant changes in H3K79 methylation upon CALM-$_{PZP}$AF10$^{MF}$ expression. BH: Benjamini Hochberg. **b** Meta-analysis of H3K79me2 centered around the transcription start site (TSS) of loci with differential H3K79me2 distribution in CALM-AF10$^{MF}$ samples (left) compared to CALM-AF10$^{MF}$ + CALM-$_{PZP}$AF10$^{MF}$ samples (blue plots) is shown above corresponding heatmaps. $Y$-axis represents genes, whereas $X$-axis shows the distance in base pair (bp) from the TSS for each gene. Peak density increases from blue to red. The left three panels and the right three panels show input and two replicates of CALM-AF10$^{MF}$ and CALM-AF10$^{MF}$ + CALM-$_{PZP}$AF10$^{MF}$ samples, respectively.

**Western blotting.** 293T cells were transfected with 1XFLAG CALM-AF10$^{MF}$ or 1XFLAG CALM-$_{PZP}$AF10$^{MF}$. Non-transfected cells were used as controls. Whole-cell lysates were prepared by lysing cells in RIPA buffer containing Protease and Phosphatase Inhibitor Cocktail (ThermoFisher Scientific). Proteins were quantified using Bradford protein assay (Bio-rad) and processed using NuPAGE LDS sample buffer and reducing reagent (ThermoFisher Scientific) for loading equal amounts (40 μg) onto the gels. SDS-PAGE electrophoresis was done using NuPAGE 4–12% Tris-glycine gels and proteins were transferred to nitrocellulose membrane using iBlot 2 gel transfer device and gel stacks (ThermoFisher Scientific). Primary antibodies were diluted in blocking buffer (4% milk in TBS-Tween20) or in 5% BSA in

TBST and incubated overnight at 4 °C. Then incubated with horseradish peroxidase (HRP)-conjugated anti-mouse or anti-rabbit secondary antibodies for an hour. Dilutions for all the antibodies are mentioned in the source data file. Blots were developed using Western ECL substrate (PerkinElmer) and images were acquired using ChemiDoc MP Imaging System (Bio-Rad) and processed using Image Lab Software (Bio-Rad).

**Flow cytometry.** 293T cells were transfected with 1XFLAG CALM-AF10$^{MF}$ or 1XFLAG CALM-$_{PZP}$AF10$^{MF}$ constructs. Forty-eight hours after transfection, cells were trypsinized, spun down, and resuspended in 500 μL PBS for flow cytometry. Sytox Blue was used as a viability stain to remove dead cells from samples during analysis. Samples were analyzed for Green Fluorescent Protein (GFP) +ve cells. Non-transfected 293T cells were used as control.

**DNA cloning and protein purification.** AF10$_{PHD1}$ (aa 20–75) and AF10$_{PZP}$ (aa 19–208) of mouse AF10 were cloned into pGEX 6p-1 and pDEST15 vectors, respectively. The Y41W and D43A mutants of AF10$_{PHD1}$ and the D43K and E179K mutants of AF10$_{PZP}$ were generated using the Stratagene QuickChange Lightning Site-Directed Mutagenesis kit. The sequences were confirmed by DNA sequencing. All proteins were expressed in *Escherichia coli* Rosetta-2 (DE3) pLysS cells grown in either Luria Broth or in minimal media supplemented with $^{15}NH_4Cl$ (Sigma) or $^{14}NH_4Cl$ (for unlabeled proteins) and $ZnCl_2$. Protein production was induced with 0.5–1.0 mM IPTG for 18 h at 16 °C. Bacteria were harvested by centrifugation and lysed by sonication in buffer (25–50 mM Tris-HCl pH 7.0–7.5, 150–500 mM NaCl, 0.05% (v/v) Nonident P 40, 5 mM dithiothreitol (DTT), and DNase). GST-fusion proteins were purified on glutathione agarose 4B beads (Thermo Fisher Sci). The GST-tag was cleaved with either PreScission or tobacco etch virus (TEV) protease. Proteins were further purified by size exclusion chromatography (SEC) and concentrated in Millipore concentrators (Millipore).

**X-ray crystallography.** For structural studies, the H3-GSGSS-AF10$_{PZP}$ construct (aa 1–12 of histone H3, a GSGSS linker, and aa 19–208 of AF10) was cloned into a pDEST15 vector with the N-terminal GST tag and TEV cleavage site. The linked protein was produced as above. Following cleavage with TEV protease and further purification by SEC, the linked H3-PZP protein was concentrated in (50 mM Tris-HCl pH 7.5, 500 mM NaCl, 5 mM DTT). Crystals were grown at 4.5 mg/ml (25 mM Tris-HCl pH 7.5, 150 mM NaCl, 5 mM DTT) using sitting-drop diffusion method at 18 °C by mixing 500 nL of protein with 500 nL of well solution composed of 90 μl (0.1 M Tris pH 8.5, 25% PEG 3350) and 10 μL 0.1 M spermine tetrahydrochloride. Crystals were cryoprotected with 30% (v/v) glycerol. X-ray diffraction data were collected at the ALS 4.2.2 beamline, Berkeley. Indexing and scaling were completed using XDS[27]. The phase solution was found using the single-wavelength anomalous dispersion method with Zn anomalous signal in phenix[28]. Model building was performed with Coot[29], and the structure was refined using phenix.refine. The final structure was validated with MolProbity[30]. The X-ray diffraction and structure refinement statistics are summarized in Supplementary Table 1.

**NMR experiments.** Nuclear magnetic resonance (NMR) experiments were performed at 298 K on Varian INOVA 900 MHz and 600 MHz spectrometers equipped with cryogenic probes. The NMR samples contained 0.1–0.2 mM uniformly $^{15}N$-labeled WT or mutated AF10$_{PHD1}$ or AF10$_{PZP}$ in either 50 mM sodium phosphate buffer pH 6.9, supplemented with 50 mM NaCl, 2 mM dithiothreitol, or 50 mM Tris-HCl pH 7.5 buffer, supplemented with 150 mM NaCl, 5 mM DTT and 8-10% $D_2O$. Binding was characterized by monitoring chemical shift changes in $^1H,^{15}N$ HSQC spectra of the proteins induced by the addition of H3 peptides (synthesized by Synpeptide) or 147 bp 601 Widom DNA. NMR data were processed and analyzed as previously described[31].

Uniformly $^{15}N$-labeled histone H3 (aa 1–44) was expressed and purified as described previously[32]. The protein was purified over several columns and lyophilized. The NMR sample contained 0.05 mM $^{15}N$-labeled H3$_{1-44}$ in 20 mM MOPS pH 7.0, 150 mM KCl and 1 mM DTT. Binding was monitored as above, upon the addition of unlabeled WT AF10$_{PZP}$.

**Fluorescence spectroscopy.** Spectra were recorded at 25 °C on a Fluoromax-3 spectrofluorometer (HORIBA) as described previously[33] with the following modifications. The samples containing 0.5–1 μM wild-type or mutant AF10$_{PZP}$ or AF10$_{PHD1}$ and progressively increasing concentrations of H3 (1–12, 15–34, and 1–13 aa) peptides were excited at 295 nm. All experiments were performed in buffer containing 50 mM Tris-HCl pH 7.5, 150 mM NaCl, 5 mM DTT. Emission spectra were recorded over a range of wavelengths between 310 and 380 nm with a 0.5 nm step size and a 1 s integration time. The $K_d$ values were determined using nonlinear least-squares analysis and the equation:

$$\Delta I = \Delta I_{max}\left(([L] + [P] + K_d) - \sqrt{([L] + [P] + K_d)^2 - 4[P][L]}\right)/2[P] \quad (1)$$

where $[L]$ is the concentration of the histone peptide, $[P]$ is the protein concentration, $\Delta I$ is the observed change of signal intensity, and $\Delta I_{max}$ is the difference

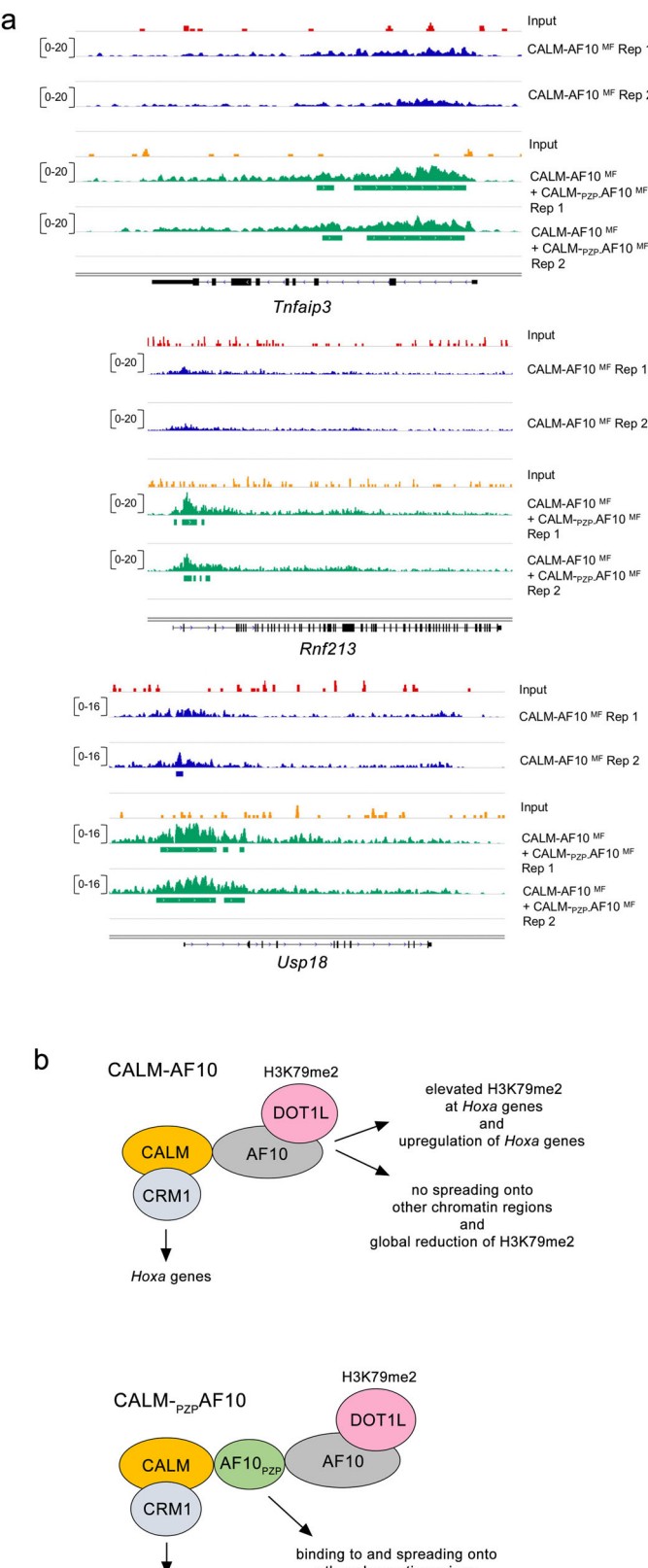

**Fig. 10 The role of AF10$_{PZP}$ in CALM-AF10-mediated leukemogenesis. a** Representative profiles of H3K79me2 peaks in CALM-AF10$^{MF}$ leukemias without (upper panels, blue tracks) or with co-transduction of the CALM-$_{PZP}$AF10$^{MF}$ fusion (lower panels, green tracks). Red tracks and orange tracks represent ChIP input control for CALM-AF10$^{MF}$ and CALM-AF10$^{MF}$ + CALM-$_{PZP}$AF10$^{MF}$ samples, respectively. Data from 2 independent ChIP replicates is shown. **b** Schematic of the mechanism of AF10$_{PZP}$-dependent CALM-AF10-mediated leukemogenesis.

in signal intensity of the free and bound states of the domain. The $K_d$ values were averaged over three separate experiments, with the error calculated as standard deviation between the runs.

**Peptide pull-down assay**. One microgram of biotinylated histone peptides with different modifications was incubated with 1 µg of GST-AF10$_{PZP}$ in binding buffer (50 mM Tris-HCl pH 7.5, 250 mM NaCl, 0.1% NP-40, and 1 mM PMSF) overnight. Streptavidin magnetic beads (Pierce) were added to the mixture, and the mixture was incubated for 1 h with rotation. The beads were then washed three times and analyzed using SDS-PAGE and western blotting.

**Nucleosome preparation**. Human H2A, H2B, H3.2, and H4 histone proteins were expressed in *Escherichia coli* BL21 DE3 pLysS cells, separated from inclusion bodies, and purified using size exclusion and ion-exchange chromatography, as described previously[34]. Histones were then mixed together in 7 M guanidine HCL, 20 mM Tris-HCl pH 7.5, and 10 mM dithiothreitol in appropriate molar ratios and refolded into octamer by slow dialysis into 2 M NaCl, 20 mM Tris-HCl pH 7.5, 1 mM ethylenediaminetetraacetic acid (EDTA) pH 8.0, and 2 mM β mercaptoethanol. The octamer was purified from tetramer and dimer by size exclusion chromatography. Octamer was then mixed with 20% excess of DNA in 2 M NaCl, 5 mM Tris pH 8.0, and 0.5 mM EDTA pH 8.0, and NCPs were reconstituted from octamer plus DNA by slow desalting dialysis into 5 mM Tris pH 8.0 and 0.5 mM EDTA pH 8.0. Finally, the NCPs were separated from free DNA via sucrose gradient purification. DNAs used were either the 147 bp 601 Widom NPS flanked with 30 bp linker DNA on either side and internally labeled with fluorescein 27 bp in from the 5' end, or the 601 Widom NPS labeled with fluorescein or Cy3 on the 5' end.

**Fluorescence polarization**. Fluorescence polarization measurements were carried out by mixing increasing amounts of AF10 WT or D43A and E179K mutants with 5 nM NCPs in 75 mM NaCl, 25 mM Tris-HCl pH 7.5, 0.00625% Tween20, and 5 mM dithiothreitol in a 30 µL reaction volume. The samples were loaded into a Corning round bottom polystyrene plate and polarization measurements were acquired with a Tecan infinite M1000Pro plate reader by exciting at 470 nm and measuring polarized emission at 519 nm with 5 nm excitation and emission bandwidths. The fluorescence polarization was calculated from the emission polarized parallel and perpendicular to the polarized excitation light as described previously[35]. The data were then fit to a binding isotherm to determine $S_{1/2}$s. The $S_{1/2}$ values were averaged over three separate experiments with the error calculated as the standard deviation between the runs.

**EMSA**. EMSAs were performed by mixing increasing amounts of AF10$_{PZP}$ with 0.25 pmol of 601 Widom DNA/lane in 20 mM Tris-HCl pH 7.5 buffer supplemented with 150 mM NaCl and 5 mM dithiothreitol in a 10 µL reaction volume. Reaction mixtures were incubated at 4 °C for 10 min and loaded onto a 5% native polyacrylamide gel. Electrophoresis was performed in 0.2× Tris-borate-EDTA (TBE) at 80–100 V on ice. The gels were stained with SYBR Gold (Thermo Fisher Sci) and visualized by Blue LED (UltraThin LED Illuminator-GelCompany). EMSAs with NCPs were performed by mixing increasing amounts of AF10$_{PZP}$ WT or D43A and E179K mutants with 5 nM NCP$_{207}$ in 75 mM NaCl, 25 mM Tris-HCl pH 7.5, 10% glycerol, and 0.005% Tween 20 buffer in a 12 µL reaction volume. Each sample was incubated at 4 °C for 5 min and then loaded onto a 5% native polyacrylamide gel. Electrophoresis was performed in 0.3× Tris-borate-EDTA (TBE) at 300 V for 90 min. Fluorescence images were acquired with a Typhoon Phosphor Imager.

**FRET**. FRET efficiency measurements were carried out on a Horiba Scientific Fluoromax 4. The data were collected using FluorEssence v3.5 software and processed with Matlab R201a. Samples were excited at 510 and 610 nm and the photoluminescence spectra were measured from 530 to 750 nm and 630 to 750 nm for donor and acceptor excitations, respectively. Each wavelength was integrated for one second, and the excitation and emission slit width was set to 5 nm with 2 nm emission wavelength steps. FRET efficiencies were computed through the (ratio)A method[36]. AF10 titrations were carried out in 75 mM NaCl, 25 mM Tris-HCl pH 7.5, 0.00625% Tween20, 10% glycerol, and 5 mM dithiothreitol with 5 nM nucleosomes in a 20 µL reaction volume.

**H3K79me2 ChIP-seq data analysis**. Adapter remnants of sequencing reads were removed with cutadapt v2.3[37]. Trimmed ChIP-seq sequencing reads were aligned to mouse genome version 38 (mm10) using STAR aligner version 2.7[38]. ChIP-seq reads and alignment quality was assessed using FastQC v0.11.5. Homer v4.10 was used to call peaks from ChIP-seq samples, annotate peaks to mouse genes, and quantify reads count to peaks. Ensembl gene annotations version 84 were used in the alignment and quantification steps. The raw read count for different peaks was compared using DESeq2 v1.22.2[39] based on a generalized linear model. Peaks with a Benjamini-Hochberg adjusted $P$ value <0.05 and fold change ≥1.5 or ≤0.6667 were selected as significantly differentially marked (DM) peaks. Genes associated with any DM peaks at exon, intron, promoter, transcription termination site, and

closest intergenic region were investigated for GO and pathway functional enrichment tests using Ingenuity Pathway Analysis (Qiagen, Redwood City, USA). To view the H3K79me2 changes for DM peaks associated genes, we generated normalized signal density profiles over the TSS +/− 10 kb using deepTools v3.5.1.

## Data availability

Coordinates and structure factors for H3$_{1-12}$-AF10$_{PZP}$ have been deposited in the Protein Data Bank with PDB ID 7MJU. The ChIP-seq data generated in this study are available on GEO under the accession number GSE163170. For AF10 fusion analysis, publicly available data from the St. Jude server were used (https://pecan.stjude.cloud/). All other relevant data supporting the key findings of this study are available within the article and its Supplementary Information file or from the corresponding authors upon reasonable request. Source data are provided with this paper.

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

## Acknowledgements

We thank Jay Nix for assistance with the collection and processing of X-ray diffraction data, Jaewoo Ahn for the DNA construct design, and Brian James and Yoav Altman for next-generation sequencing and flow cytometry respectively. We also acknowledge the Functional Genomics core at SBP Medical Discovery Institute for preparing viral supernatants for this study. This work was supported by grants from NIH HL151334, CA252707, AG067664, GM125195 and GM135671 to T.G.K. and GM120582, GM121966 and GM131626 to M.G.P. A.J.D. would like to acknowledge support from the NIH grants CA154880 and NIH/NCI CA030199, the Rally Foundation for Childhood Cancer Research and Luke Tatsu Johnson Foundation Award (19YIN45), an Emerging Scientist Award from the Children's Cancer Research Fund, and the V Foundation for Cancer Research Award (DVP2019-015). A.D. would like to acknowledge support from the Lady Tata foundation.

## Author contributions

B.J.K., A.D., K.L.C., F.X., M.Z., K.B., S.K., Q.T., Y.Z., P.Z. and A.S. performed experiments and together with S.K.B., X.S., H.W., M.G.P., A.J.D. and T.G.K. analyzed the data. A.J.D. and T.G.K. wrote the manuscript with input from all authors.

## Competing interests

The authors declare no competing interests.
