## [Peer Review File · Nature Communications]

REVIEWER COMMENTS

Reviewer #1 (Remarks to the Author):

The Klein et al study identifies a role for the AF10 PZP domain in blocking CALM-AF10 fusion-driven leukemogenesis. CALM-AF10 fusions lack at least the first PHD finger of the PZP module if not the entire module, suggesting that the molecular activity of this motif may inhibit CALM-AF10 fusion pathologic functions. The authors convincingly show that inclusion of the PZP domain in a transforming CALM-AF10 fusion construct can significantly inhibit the transformative properties of the fusion protein in cells and in in vivo models. Importantly, the authors show that the inhibitory activity of the PZP domain may work in trans as well. The AF10 PZP domain was previously shown to bind to nucleosomes through recognition of unmodified H3K27. In this study, the authors perform a rigorous and detailed biophysical and structural analysis of the histone/nucleosome-binding activities for the first PHD finger of AF10 and for the entire PZP domain. Interestingly, the authors find that the module can use a different mode to bind to the N-terminal region of H3 independent of the previously identified K27 region. Structure-guided mutants are identified that separate the two binding activities of the PZP domain, providing convincing functional support for the biophysical findings. Finally, ChIP-seq studies show that the AF10 PZP domain chromatin localization is excluded from H3K27me3-enriched regions in cells but is not impacted by H3K4me3, providing a compelling model for how this module can be tumor suppressive in the context of the CALM-AF10 fusion.

Overall, this is an interesting and important multi-disciplinary study that should be of broad interest across many fields. I have a couple of comments for the authors to consider.

Comments:

1) In Figures 1 and 2: Cell types likely make it difficult, but it would be helpful to have some information about the expression levels for the two constructs. One possibility is to express the ectopic constructs in a standard cell line to test whether the addition of the PZP domain into the CALM-AF10mf is tolerated or if it causes the protein to misfold, which would be an alternative explanation of the results.

2) Given the elegant studies identifying point mutants that separate the different binding activities of the PZP domain (H3 1-10 vs sequence spanning K27), is there a straightforward manner to functionally test the different biochemical activities? I should add that it would not be fair to ask - particularly given the situation with COVID - for the authors to repeat all of the many difficult functional studies in the paper with the different mutants. If there is not a straightforward way of addressing the request, adding within the discussion the authors' views for how the two activities may work in the context of AF10 physiology/pathology would be helpful.

Reviewer #2 (Remarks to the Author):

It has been proposed that CALM-AF10-mediated mis-localization of DOT1L to chromatin causes changes in H3K79 methylation and gene expression and contributes to leukemic transformation. However the precise mechanism by which DOT1L is mis-localized remains unclear. An almost entire CALM protein is fused to AF10, in which the first PHD finger is deleted, results the CALM-AF10 chimera, however whether impaired first PHD affects the transforming ability of AF10 fusions is unknown.

In this study, the authors describe the biological function of the N-terminal PHD1-zinc-knuckle-PHD2 (PZP) domain of AF10 (AF10PZP) and its critical role in inhibiting leukemogenic activity of the CALM-AF10. They report the molecular mechanism by which AF10PZP recognizes a large portion of the histone H3 tail and DNA and assess the contribution of these binding events. Their data suggest that the disruption of AF10PZP function in oncogenic AF10 fusions leads to malignant transformation, whereas inclusion of AF10PZP reverses leukemogenesis.

The function of the first PHD finger of AF10 was unknown in the past. From the functional and subsequent

structural analysis, the authors revealed that it was possible to explain why this area, the N-terminal PHD1-zinc-knuckle-PHD2 (PZP) of AF10, disappeared in leukemogenic CALM-AF10. Observed findings contain very interesting and presented results seem to be sound. The experiments were well designed. This paper was a joy to read!

However this reviewer still has some concerns.

1) In Fig. 2 c. What is the trans-mechanism by which the introduction of CALM-PZPAF10MF suppresses the function of leukemia cells generated by CALM-AF10MF? This reviewer can understand individual function in leukemogenic CALM-AF10MF and non-leukemogenic CALM-PAPAF10MF, however still cannot understand trans- inhibition mechanism by CALM-PZPAF10MF on CALM-AF10MF. How about the quantitative ratio between CALM-PZPAF10MF and CALM-AF10MF? Was CALM-PZPAF10MF over-produced?

Previous reports (Blood. 2013, 121:4758-68; Cancer Sci. 2014, 105:315-23.) have stated that CALM-AF10 is mainly present in the cytoplasm because CALM has NES, and part of it is recruited to Hox, Meis1 etc. From these reports, one can imagine two possibilities.

i) Since CALM-AF10MF is released to the outside of the nucleus, the original function of AF10 portion cannot be achieved, resulting in leukemia. CALM-PZPAF10MF can rescue above situation because it remains in the nucleus and fulfills the original function of AF10 portion.

ii) CALM-PZPAF10MF inhibit chromatin binding of CALM-AF10MF in a competitive manner or by other mechanisms.

In line 324-326, authors wrote that "Incorporation of the chromatin reader, AF10PZP in the CALM-AF10 fusion allows for spreading onto other regions of chromatin, thus disseminating DOT1L throughout the genome", this reviewer assumes that authors may have different model. Please explain.

2) Related to the above point, CALM-PZPAF10MF and CALM-AF10MF lack NLS of AF10 and retain NES of CALM. How about cellular localization of CALM-PZPAF10MF and CALM-AF10MF? Since in Fig. 8, authors added 2XNLS to AF10PZP and did ChIP-seq, this suggest PZP has no nuclear localization ability. I think the information about the localization and function of AF10, which lacks PZP, will contribute to understanding the function of AF10.

3) This reviewer wonder if the effect of PZP is valid only when the shortened CALM-PZPAF10MF and CALM-AF10MF were used here or not. It remains a question whether the same inhibitory effect can be seen by introducing full length CALM-PZPAF10 into the leukemia cells generated by original CALM-AF10 used in previous report (Blood. 2013, 121:4758-68; Cancer Sci. 2014, 105:315-23.) or CALM-AF10MF used in this study.

4) In many cases CALM-AF10 lacks PHD1 and it appears that PHD1 alone is sufficient to interact with H3 in Fig.3. The authors clearly revealed that PHD1 and Zn-kn are important domain for H3 binding and identified critical amino acid residues mutation E179K and D43K. Since in line 239-240, authors wrote that "The double D43K/E179K 240 mutation in both sites of AF10PZP is required to eliminate the interaction with H3 (Fig. 6i, j)", this reviewer just wonders whether the function of CALM-PZPAF10MF in leukemogenesis is impaired by these single or double mutations? If mutations are indeed affect the function of CALM-PZPAF10MF in leukemogenesis, the results would greatly improve the impact of this report.

5) It has been reported that in Drosophila a short splicing form that does not contain PHD1, equivalent to PZP, is functional (Mol Genet Genomics. 2004 Sep;272(2):156-61.). This report suggests that PHD1 is not so important for AF10 function at least in Drosophila. I understand that it is different from the context of CALM-AF10, but are there any comments for this result? This reviewer would like to hear authors' comments if possible.

6) AF9 and ENL are also known as DOT1L binding protein. They have OM-LZ domain and YEAT as histone binding domain. They also form leukemogenic fusion genes as MLL-AF9 or MLL-ENL. AF10 also form leukemogenic fusion genes as MLL-AF10. Is it possible that the YEAT has the same function as the PZP of

AF10 and it serve same function of PZP even in the fusion genes? If possible, I would like to hear authors' opinion.

Minor points

1) Styles of Fig. 2 c and Fig. 8 d should be re-considered. If I understand correctly, in these experiments, MIG (no insert) or CALM-PZPAF10MF were transduced into CALM-AF10MF leukemia cells, however I first mis-understood that cells were transduced CALM-AF10MF or CALM-PZPAF10MF from present style. Please reconsider.

2) MOLM13 cells has MLL-AF9 translocation. This reviewer just wonders why authors choose this particular cell line for experiments shown in Fig. 8.

3) On my PC, "DTT" in line 416, 417, "ALS 4.2.2" in line 420 and "The X-ray diffraction and structure refinement statistics are summarized in Supplementary Table 1" in line 424-425, show strange color of letters. Please check them.

4) Why authors did not tach with results of H21-44 in Fig. 3?

5) In line 520, please change XXX if these are fixed.

Reviewer #3 (Remarks to the Author):

Klein et al. showed that the PZP domain of AF10 (AF10PZP) inhibits transforming activity of the leukemia-associated CALM-AF10 fusion. The authors also demonstrated that AF10PZP contacts with histone H3 tail and DNA to associate with chromatin. This finding is consistent with the previous report (ref 15), which shows that AF10PZP senses unmodified H3K27 to regulate DOT1L-mediated methylation of H3K79. However, the physiological significance of the inhibition of CALM-AF10 by AF10PZP is still unclear.

1. To test the role of AF10PZP in leukemogenic activity, AF10PZP was incorporated into a small/artificial version of CALM-AF10. Original CALM-AF10 fusion, which is found in patients, should be used.

2. The authors claim that CAL-AF10 with PZP may bind to and spread onto other chromatin regions. To compare genome distribution of CALM-AF10 and CALM-AF10 containing PZP, ChIP-sequence analyses of these fusions should be performed.

3. Does AF10PZP inhibit other leukemic fusions such as MLL-AF10?

4. The AF10PZP mutants such as D43K and E179K were defective in binding to the H3. Do these mutations reverse transforming activity of CAL-AF10 containing PZP?

Reviewer #4 (Remarks to the Author):

AF10 translocations are frequently found associated with human leukemia. In this paper, the authors found that inclusion of the PZP domain of AF10 in the CALM-AF10 fusion protein prevented the transforming activity of HSP cells, and abrogated CALM-AF10 mediated leukemia in mouse models. The authors proceeded to explore the molecular mechanism of AF10 PZP, through crystallography, NMR and binding analysis, they found that the PZP domain use two different surfaces to recognize both the 1-6 of and 21-27 fragments of H3, thus providing a model of AF10 recognizing the nucleosome through extensive H3 and DNA contacts. In this study, the authors identified an unexpected inhibitory role of the PZP domain in CALM-AF10 mediated leukemia, which is very important in understanding AF10-fusion related leukemia. The authors proceed to identify the histone recognition feature of AF10 PZP, which revealed new findings and complemented

previous studies in AF10 PZP. So the molecular mechanism presented in this paper should represent an important progress in understanding AF10-related leukemia.

Several minor points related to this paper should be noted:

1. E179K mutation is mentioned several times in this paper, but the position of this mutation is not shown. The authors may need to show this residue somewhere to facilitate understanding of related analysis in this paper.
2. In figure 4b and 4c, the authors presented several water mediated hydrogen bonds. Water mediated hydrogen bonds may be due to crystal packing. If the authors can not prove the importance of those waters, it's better not to mention about those interactions.
3. In all the HSQC spectra in supplementary figures 2-10, the x and y axes should be properly labelled.

We would like to thank the Reviewers for the insightful comments which were very helpful in revising and strengthening this manuscript.

New results are shown in Figs. 2d, 8a, 9a, 9b, and 10a and Suppl. Figs. 1a and 1b.

Reviewer 1:

1) In Figures 1 and 2: Cell types likely make it difficult, but it would be helpful to have some information about the expression levels for the two constructs. One possibility is to express the ectopic constructs in a standard cell line to test whether the addition of the PZP domain into the CALM-AF10mf is tolerated or if it causes the protein to misfold, which would be an alternative explanation of the results.

Response: We would like to thank the reviewer for this suggestion. The expression level for two constructs (shown in new Suppl. Fig. 1 a and b) suggests that incorporation of the PZP domain into CALM-AF10^{MF} is tolerated.

2) Given the elegant studies identifying point mutants that separate the different binding activities of the PZP domain (H3 1-10 vs sequence spanning K27), is there a straightforward manner to functionally test the different biochemical activities? I should add that it would not be fair to ask - particularly given the situation with COVID - for the authors to repeat all of the many difficult functional studies in the paper with the different mutants. If there is not a straightforward way of addressing the request, adding within the discussion the authors' views for how the two activities may work in the context of AF10 physiology/pathology would be helpful.

Response: The difference between binding affinities of PZP for each H3 site and both sites (synergistic binding, Fig. 5) may not provide straightforwardly detectable differences in functional assays. We used instead the double D43K/E179K mutant to define the role of PZP in sub-cellular distribution of the fusion (new Fig. 8a). We found that inclusion of the functional PZP domain (but not the D43K/E179K mutant) shifts the localization of CALM-AF10 from predominantly cytoplasmic to nuclear. Cytoplasmic localization of the CALM-AF10 fusion protein has been shown to be essential for its leukemogenic activity (Conway AE et al., Blood, 2013, ref. #23). In support, our results demonstrate the importance of AF10 PZP loss in ensuring cytoplasmic localization of the oncogenic CALM-AF10 fusion.

Reviewer 2:

1) In Fig. 2 c. What is the trans-mechanism by which the introduction of CALM-PZPAF10MF suppresses the function of leukemia cells generated by CALM-AF10MF? This reviewer can understand individual function in leukemogenic CALM-AF10MF and non-leukemogenic CALM-PAPAF10MF, however still cannot understand trans- inhibition mechanism by CALM-PZPAF10MF on CALM-AF10MF. How about the quantitative ratio between CALM-PZPAF10MF and CALM-AF10MF? Was CALM-PZPAF10MF over-produced?

Response: The data shown in new Suppl. Fig. 1 reveal that CALM-PZP-AF10 MF is not over-produced compared to CALM-AF10. Overexpression of DOT1L or AF10 in leukemia cells has been found to lead to spreading of H3K79me2 on genes other than leukemia targets and resulting in loss of leukemic proliferation (Kingsley MC et al., Blood Advances 2020, ref. #25). In support, our data indicate that expression of the CALM-PZP-AF10MF in CALM-AF10 MF leukemia cells also leads to spreading of H3K79me2 (see new Fig. 9 and 10).

Previous reports (Blood. 2013, 121:4758-68; Cancer Sci. 2014, 105:315-23.) have stated that CALM-AF10 is mainly present in the cytoplasm because CALM has NES, and part of it is recruited to Hox, Meis1 etc. From these reports, one can imagine two possibilities.

- i) Since CALM-AF10MF is released to the outside of the nucleus, the original function of AF10 portion cannot be achieved, resulting in leukemia. CALM-PZPAF10MF can rescue above situation because it remains in the nucleus and fulfills the original function of AF10 portion.
- ii) CALM-PZPAF10MF inhibit chromatin binding of CALM-AF10MF in a competitive manner or by other mechanisms.

Response: We thank the reviewer for pointing this out. We now show indeed that CALM-AF10 is predominantly cytoplasmic, while inclusion of PZP promotes nuclear localization (new Fig. 8a). Just as the reviewer suggested, it is therefore likely that PZP loss is important for abrogating the original function of AF10. Interestingly, we further show that mutating residues in the PZP domain that abolish H3 tail binding reverses the nuclear localization of the CALM-PZP-AF10 fusion, indicating the importance of functional PZP in this process. We also cite the suggested references Blood. 2013, 121:4758-68; Cancer Sci. 2014, 105:315-23, refs. #23, 24.

In line 324-326, authors wrote that “Incorporation of the chromatin reader, AF10PZP in the CALM-AF10 fusion allows for spreading onto other regions of chromatin, thus disseminating DOT1L throughout the genome”, this reviewer assumes that authors may have different model. Please explain.

Response: In our first submission, we had proposed that PZP incorporation may promote spreading of H3K79me2 to other regions (non-CALM-AF10 target genes) across the genome. We now tested this hypothesis, and we see indeed gain of H3K79me2 at several novel loci after CALM-PZP-AF10 expression in CALM-AF10 leukemia cells (new Figs. 9-10).

2) Related to the above point, CALM-PZPAF10MF and CALM-AF10MF lack NLS of AF10 and retain NES of CALM. How about cellular localization of CALM-PZPAF10MF and CALM-AF10MF? Since in Fig. 8, authors added 2XNLS to AF10PZP and did CHIP-seq, this suggest PZP has no nuclear localization ability. I think the information about the localization and function of AF10, which lacks PZP, will contribute to understanding the function of AF10.

Response: As mentioned above, our new immunofluorescence studies actually show that the PZP of AF10 is sufficient to completely change the localization of CALM-AF10^{MF} from cytoplasmic to nuclear despite the fact that these CALM-AF10 minimal fusions do not contain the AF10 NLS sequence. It is important to note that the full-length CALM-AF10 fusion is also predominantly cytoplasmic (Lin YH Blood 2009, ref. #14), implying that the AF10 NLS does not dictate sub-cellular localization in the context of the CALM-AF10 fusion protein.

3) This reviewer wonder if the effect of PZP is valid only when the shortened CALM-PZPAF10MF and CALM-AF10MF were used here or not. It remains a question whether the same inhibitory effect can be seen by introducing full length CALM-PZPAF10 into the leukemia cells generated by original CALM-AF10 used in previous report (Blood. 2013, 121:4758-68; Cancer Sci. 2014, 105:315-23.) or CALM-AF10MF used in this study.

Response: We agree with the Reviewer. In order to address this important question, we cloned the full length AF10 protein (all 1027 amino acids, including the PZP domain)

downstream of CALM. We compared the activity of this CALM-full-AF10 fusion to the separately cloned CALM-AF10 fusion protein found in U937 cells. Even using these fusions (not minimal fusions), our results show that while CALM-AF10 could potentially activate *Hoxa/Meis1* genes in mouse bone-marrow derived hematopoietic stem and progenitor cells (HSPCs), the CALM-full-AF10 fusion failed to do so (see new Fig. 2d). These results reiterate the notion that loss of the AF10 N-terminal PZP domain, which is absent in the U937 derived CALM-AF10 fusion, is important for its leukemic activity, which we have studied in detail using the more easy-to-use minimal fusion proteins.

4) In many cases CALM-AF10 lacks PHD1 and it appears that PHD1 alone is sufficient to interact with H3 in Fig.3. The authors clearly revealed that PHD1 and Zn-kn are important domain for H3 binding and identified critical amino acid residues mutation E179K and D43K. Since in line 239-240, authors wrote that “The double D43K/E179K 240 mutation in both sites of AF10PZP is required to eliminate the interaction with H3 (Fig. 6i, j)”, this reviewer just wonders whether the function of CALM-PZPAF10MF in leukemogenesis is impaired by these single or double mutations? If mutations are indeed affect the function of CALM-PZPAF10MF in leukemogenesis, the results would greatly improve the impact of this report.

Response: While PHD1 binds to the H3 sequence (aa 1-6), the synergistic binding of PHD1 to H3(1-6) and PHD2/Zn-kn to H3(21-27) is necessary for the strong interaction of PZP with H3 (see K_d values in Fig. 5e). D43 is a residue of PHD1 and E179 is a residue of PHD2. As mentioned previously, we have now shown in cells that incorporation of PZP promotes changes in localization of the CALM-AF10 fusion from cytoplasmic to nuclear, which is reversed upon introduction of D43K/E179K mutations in PZP. Note, that the NMR data show that neither mutation disrupts the structure of PZP. Although we haven't done experiments with single point mutants, we expect that the effect of a single mutation on promoting the nuclear localization will be weaker compared to the effect of the double mutation (compare binding affinities for histone H3 (1-31aa): of WT, 0.3 μ M; D43K, 2.9 μ M; and E179K, 7.8 μ M). See also response to comment 2 of Reviewer 1.

5) It has been reported that in *Drosophila* a short splicing form that does not contain PHD1, equivalent to PZP, is functional (Mol Genet Genomics. 2004 Sep;272(2):156-61.). This report suggests that PHD1 is not so important for AF10 function at least in *Drosophila*. I understand that it is different from the context of CALM-AF10, but are there any comments for this result? This reviewer would like to hear authors' comments if possible.

Response: In fact, in Drosophila, the study cited above and a prior study by the same group Perrin et al., Mol Cell Biol. 2003) showed that deletion of the AF10 PZP equivalent in Drosophila activates homeotic genes and leads to homeotic defects, whereas inclusion of PZP reverses those effects. Effectively, these studies in Drosophila show that PZP exclusion is required for homeotic i.e. HOX gene activation by AF10 during body patterning - further highlighting the importance of AF10 PZP for homeotic gene regulation across organisms.

6) AF9 and ENL are also known as DOT1L binding protein. They have OM-LZ domain and YEAT as histone binding domain. They also form leukemogenic fusion genes as MLL-AF9 or MLL-ENL. AF10 also form leukemogenic fusion genes as MLL-AF10. Is it possible that the YEAT has the same function as the PZP of AF10 and it serve same function of PZP even in the fusion genes? If possible, I would like to hear authors' opinion.

Response: This is indeed true, and we have been intrigued that AF9 and ENL YEATS domains are disrupted in MLL-ENL and MLL-AF9 fusions. It does seem that loss of

chromatin reading and thus normal function of the proteins (AF10, ENL or AF9) may be a common mechanism of transformation in leukemia.

Minor points

1) Styles of Fig. 2 c and Fig. 8 d should be re-considered. If I understand correctly, in these experiments, MIG (no insert) or CALM-PZPAF10MF were transduced into CALM-AF10MF leukemia cells, however I first mis-understood that cells were transduced CALM-AF10MF or CALM-PZPAF10MF from present style. Please reconsider.

Response: We agree and have changed the labels to CALM-AF10^{MF} and CALM-AF10MF+CALM-PZPAF10^{MF} in the figure to make it clear.

2) MOLM13 cells has MLL-AF9 translocation. This reviewer just wonders why authors choose this particular cell line for experiments shown in Fig. 8.

Response: We performed ChIP-seq for AF10 in MOLM13 cells because we had already performed extensive characterization of several other chromatin marks in this cell line. The purpose of these experiments was simply to assess the localization of the AF10 PZP domain and its relationship to other chromatin marks. For looking at the effect of H3K79me changes upon CALM-AF10 and CALM-PZP-AF10 expression, we have used CALM-AF10 primary leukemia cells as a more appropriate system.

3) On my PC, “DTT” in line 416, 417, “ALS 4.2.2” in line 420 and “The X-ray diffraction and structure refinement statistics are summarized in Supplementary Table 1” in line 424-425, show strange color of letters. Please check them. – *we have checked them.*

4) Why authors did not tach with results of H21-44 in Fig. 3? – *the structural data and measurements of K_d s for mutants (Fig. 6) show that the first PHD finger (PHD1) is responsible for binding to the H3 sequence (aa 1-6), whereas the H3 sequence (aa 21-27) is bound by PHD2/Zn-kn.*

5) In line 520, please change XXX if these are fixed. – *these have been fixed.*

Reviewer 3

1. To test the role of AF10PZP in leukemogenic activity, AF10PZP was incorporated into a small/artificial version of CALM-AF10. Original CALM-AF10 fusion, which is found in patients, should be used.

Response: We have now used the original CALM-AF10 fusion and a CALM-full-AF10 version, which contains the intact PZP, to show abrogation of Hoxa/Meis1 gene activation (see new Fig 2d). See also more detailed response above (comment 3 of Reviewer 2).

2. The authors claim that CAL-AF10 with PZP may bind to and spread onto other chromatin regions. To compare genome distribution of CALM-AF10 and CALM-AF10 containing PZP, ChIP-sequence analyses of these fusions should be performed.

Response: We and others in the field have made a number of attempts to perform direct ChIP studies to assess genome-wide CALM-AF10 binding sites. These studies may be hampered by the predominantly cytoplasmic localization of CALM-AF10. Thus, to definitively test our hypothesis that PZP incorporation may promote spreading of

H3K79me2 to other regions (non-CALM-AF10 target genes) across the genome, we performed ChIP-seq for H3K79me2 and indeed observed gain of H3K79me2 at several novel loci after CALM-PZP-AF10 expression in CALM-AF10 leukemia cells (see new Figs. 9-10).

3. Does AF10PZP inhibit other leukemic fusions such as MLL-AF10?

Response: It is possible that other chromatin reading domains in proteins, for example AF10, AF17, ENL and AF9, may also inhibit transformation when incorporated in their respective oncoprotein fusions such as MLL-AF10/17, ENL/AF9 etc, but this idea hasn't been tested.

4. The AF10PZP mutants such as D43K and E179K were defective in binding to the H3. Do these mutations reverse transforming activity of CAL-AF10 containing PZP?

Response: See response to comment 2 of Reviewer 1.

Reviewer 4

Several minor points related to this paper should be noted:

1. E179K mutation is mentioned several times in this paper, but the position of this mutation is not shown. The authors may need to show this residue somewhere to facilitate understanding of related analysis in this paper. – *both E179 and D43 residues are now depicted in Fig. 5a.*

2. In figure 4b and 4c, the authors presented several water mediated hydrogen bonds. Water mediated hydrogen bonds may be due to crystal packing. If the authors can not prove the importance of those waters, it's better not to mention about those interactions. – *as suggested, the water mediated hydrogen bonds are no longer mentioned.*

3. In all the HSQC spectra in supplementary figures 2-10, the x and y axes should be properly labelled. – *these figures have been edited as suggested.*

REVIEWERS' COMMENTS

Reviewer #1 (Remarks to the Author):

The authors have addressed my original concerns and I recommend publication.

Reviewer #2 (Remarks to the Author):

I satisfied and agreed with all authors' responses.

The original manuscript has been significantly improved with the help of meaningful recommendations by the reviewers.

I have no concerns about the manuscript.

Reviewer #3 (Remarks to the Author):

The authors answered all the criticisms raised. They showed that AF10 PZP alters cytoplasmic localization of CALM-AF10 to nucleoplasm (Figure 8a). Since this data is very important, it should be mentioned in Abstract. In addition to Conway et.al. (Blood, 2013), Suzuki et al. (Cancer Science, 2014) that demonstrates critical role of cytoplasmic localization of CALM-AF10 should be cited.